# Weather, Land and Crops in the Indus Village Model: A Simulation Framework for Crop Dynamics under Environmental Variability and Climate Change in the Indus Civilisation

Andreas Angourakis [1,2,*], Jennifer Bates [3], Jean-Philippe Baudouin [4], Alena Giesche [5], Joanna R. Walker [6], M. Cemre Ustunkaya [2], Nathan Wright [7], Ravindra Nath Singh [8], Cameron A. Petrie [2,6,*]

[1] Institut für Archäologische Wissenschaften, Ruhr Universität Bochum, Am Bergbaumuseum 31, 44791 Bochum, Germany

[2] McDonald Institute for Archaeological Research, University of Cambridge, Downing Street, Cambridge CB2 3DZ, UK; mcu22@cantab.ac.uk

[3] Department of Archaeology and Art History, Seoul National University, 1 Gwanak-ro, Gwanak-gu, Seoul 08826, Korea; jbates01@snu.ac.kr

[4] Palaeoclimate Dynamics and Variability, Geo- und Umweltforschungszentrum (GUZ), Universität Tübingen, Geschwister-Scholl-Platz, 72074 Tübingen, Germany; baudouin.jeanphilippe@gmail.com

[5] Department of Geology, Colby College, Mayflower Hill Drive, Waterville, ME 04901, USA; agiesche@colby.edu

[6] Department of Archaeology, University of Cambridge, Downing Street, Cambridge CB2 3DZ, UK; joannaruthwalker@gmail.com

[7] Department of Archaeology, University of New England, Armidale, NSW 2350, Australia; nwrigh28@une.edu.au

[8] Department of AIHC and Archaeology, Banaras Hindu University, Varanasi 221005, Uttar Pradesh, India; drravindransingh@gmail.com

[*] Correspondence: andreas.angourakis@ruhr-uni-bochum.de (A.A.); cap59@cam.ac.uk (C.A.P.)

**Abstract:** The start and end of the urban phase of the Indus civilization (IC; c. 2500 to 1900 BC) are often linked with climate change, specifically regarding trends in the intensity of summer and winter precipitation and its effect on the productivity of local food economies. The Indus Village is a modular agent-based model designed as a heuristic "sandbox" to investigate how IC farmers could cope with diverse and changing environments and how climate change could impact the local and regional food production levels required for maintaining urban centers. The complete model includes dedicated submodels about weather, topography, soil properties, crop dynamics, food storage and exchange, nutrition, demography, and farming decision-making. In this paper, however, we focus on presenting the parts required for generating crop dynamics, including the submodels involved (weather, soil water, land, and crop models) and how they are combined progressively to form two integrated models (land water and land crop models). Furthermore, we describe and discuss the results of six simulation experiments, which highlight the roles of seasonality, topography, and crop diversity in understanding the potential impact of environmental variability, including climate change, in IC food economies. We conclude by discussing a broader consideration of risk and risk mitigation strategies in ancient agriculture and potential implications to the sustainability of the IC urban centres.

**Keywords:** agent-based modeling; Indus civilization; climate change; bronze age; agriculture

## 1. Introduction

The Indus civilization (IC) is one of the great civilizations of ancient Afro-Eurasia [1]. The material culture of the IC is exceptionally rich [2], and was associated with the development of the first cities in South Asia [3]. Until now, archaeologists have identified five of such cities and a multitude of smaller settlements, well distributed within the Indus

River Basin and some neighboring areas [4]. Many lines of evidence indicate that the peoples of the IC were well connected in a web of short- and long-distance exchange with contemporaneous populations in neighboring South and Central Asia, but also Iran, Oman, Mesopotamia, the Levant, Egypt, and the Eastern Mediterranean [5].

The decline of Indus urbanism has attracted considerable attention inside and outside academia. Urbanization flourished during the Mature Harappan period (c. 2600/2500–1900 BC), while by the Late Harappan period (c. 1900–1300 BC), all but one of the urban centers had been progressively abandoned, together with certain aspects of their characteristic material culture. Furthermore, the transition from Mature to Late Harappan also seems to have involved a population displacement towards the eastern hinterlands (i.e., Eastern Punjab, Haryana, and Gujarat). However, data available to date still offer an incomplete picture [6].

Among the hypotheses posed to explain the trajectory of urbanization and de-urbanization of the IC, climate change has received increasing attention. The regional environmental variability within the Indus River Basin is high. Notably, there are two well-differentiated seasonal rains (winter and summer monsoons), which are dependent on climatic processes that reach far around the globe and can vary quite unexpectedly when perceived from the human perspective [6]. Proxy evidence to date accumulates in supporting a reconstruction that suggests the urban period enjoyed stronger winter precipitation, while the transition towards the Late Harappan period and the abandonment of cities followed the onset of a decrease of winter and summer precipitation that commenced around 4.2 k BP [7–9]. It has been suggested that this process resulted in an increase in the unpredictability of rainfall over an extended period, which potentially necessitated high levels of risk mitigation [6,10].

Agriculture was the central pillar of the IC food economy, which also included many other practices, among which animal husbandry and herding were also essential [11,12]. The crops cultivated varied both between and within regions, yet generally they were ancient varieties of wheat (*Triticum sp.*), barley (*Hordeum vulgare*), various millets (e.g., *Brachiaria sp.*, *Echinochloa sp.*, *Pennisetum sp.*, *Panicum sp.*, *Setaria sp.*), sorghum (*Sorghum bicolor*), and rice (*Oryza sativa*), among others [13–18]. Previous research has highlighted the importance of seasonality and environmental diversity and variability for choosing cropping strategies, particularly if and how different approaches to multi-cropping could be used for risk management [14].

Agriculture is bound to be acutely affected by water availability, particularly in regions such as the Indus River Basin, where weather conditions vary greatly both in space and time [10,17,19,20]. Assessment of site distribution data has shown that some ancient settlements were located closer to ancient watercourses than others, and so different approaches to water were inevitably used to support farming under different regimes of water stress [6]. We further hypothesize that well-watered areas present a temptation to farmers to focus on those crops more sensitive to water stress, in the hope of minimizing the higher risk associated with them: a risk mitigation strategy that we refer to here as *hydrophile*. In contrast, we consider prioritizing multi-cropping, in any of its many forms, as the main component of a broader *polyphile* strategy, that aims at minimizing the risk associated with agriculture by means of diversification.

Such farming practices would have been particularly vulnerable to the risks entailed by climate change [6,14]. If the end of the IC is to be attributed ultimately to climate change, a central question to consider is to which degree the decline of urban and other settlements might have been directly or indirectly linked to the underperformance or failure of agricultural food production.

The TwoRains project (2015–2021, ERC H2020-648609) set out to explore these and other questions, specifically through the multidisciplinary investigation of IC sites in Haryana, NW India. The TwoRains modeling program aimed to expose the sustainability of food production regimes, especially in terms of cropping strategies, in the face of abrupt climate change. The program has led to the development of the Indus Village, a modular multi-paradigm agent-based simulation model designed to represent environmental and

human factors as a single unified system [21]. Modeling has been carried out within the spirit of creating a virtual laboratory for socioecological systems, following a two-decade-wide trend in computational archaeology [22–30]. The Indus Village is directed toward addressing how and when food production diversity is adaptive and, conversely, under which conditions, if at all, such diversity would be adverse to urbanization. The development, documentation, and exploration of a complete version of Indus Village is ongoing. However, due to its modular structure, the parts consolidated to date can be used to execute relevant experiments whose results raise several points with far-reaching implications to both past and future agricultural economies.

## 2. Materials and Methods

### 2.1. Indus Village Model: A Road Map

The development of the Indus Village model was preceded by a plan that laid out the criteria guiding all future modeling decisions. This plan defined the model scope, spatial, and temporal scales, and the essential parts and entities to be represented.

The model was set to represent explicitly an essential element of the Indus food system: the cultivation of staple food crops. As expressed by its name, Indus Village aims at representing rural, food-producing settlements, however with no fixed predefined population size. The spatial scale is kept strictly local, with an area of 25 km$^2$ or 2500 units of one hectare ($100 \times 100$ m). The total area was chosen to contain what is commonly estimated as the catchment area of agricultural settlements without motor vehicles (i.e., 5 km radius, corresponding approximately to an hour walk on a plane) [31], leaving enough room for a simulated village to change its position and distribution. The size of land units is arbitrary, chosen as a good compromise between computational costs, and adequate representation of each of the domains involved. While smaller units would have been useful to better represent, for example, hydrodynamics, we decided to keep the model scale close to the most common sizes of small and family farms, according to modern censuses worldwide (i.e., less than 5 ha) [32]. Both total area and land unit size were fixed throughout all experiments mentioned in this paper. However, these can be modified without drastic changes to the model behavior. Early in development, the model iteration or time step was set to be a day, though some processes will only happen once or a few times every year.

The focus on cereal crop-based food systems admittedly does not incorporate the full scope of IC society or the many aspects under archaeological investigation in this field. Still, the Indus Village can be classified as a relatively holistic model, and is therefore not designed to stand as the minimum model to address a single hypothesis or line of evidence. Rather, it has been developed to serve as an artificial proxy of socioecological systems centered on prehistoric agriculture, whose behavior can be interrogated through experimentation driven by testable narratives.

To define what we believe are essential elements of a local-scale food system, our approach was to reimagine an integral, yet blurry, picture of the village life in the ancient Indus River Basin based on known archaeological evidence, but also on minimum assumptions informed by parallels in space and time. These elements were later defined and reorganized as submodels, or parts of submodels (Table 1).

Currently, all Indus Village submodels are designed and, at least partially, implemented. However, the present article covers only the first set of milestones in the modeling program, involving the weather, soil water, land, and crop models, whose implementations are fully consolidated and united in the form of two integrated models (Figure 1). The Indus Village submodels are intentionally designed as modules, meaning they can function independently as models, if the required input is given. More concretely, the weather and land models are fully independent modules, with no input required, while the soil water and crop models need the input of either the weather model or a daily weather time-series dataset. Our hope is that modularity, besides enforcing greater code stability and extensibility, will also encourage the reusability of these submodels in the future. The reusability

of the Indus Village as a whole is also further potentiated since individual modules can be modified or even replaced as long as the required input–output connections are attended.

**Table 1.** Correspondence between domains and the submodels planned for the Indus Village model.

| Domain | Aspects to Model | Model |
|---|---|---|
| Climate | *Solar radiation, temperature, and precipitation* | Weather model |
| Land and soil | *Soil properties, soil and surface water dynamics, and cover including vegetation* | Soil water model, land model |
| Food production | *Crop cultivation, with a strong focus on staple cereals, and animal husbandry, supported by fishing, hunting and gathering* | Crop model, land use mechanisms |
| Population and social structure | *Individuals organized in households, i.e., the social unit of co-habitation, production, consumption, reproduction and decision-making* | Household agent set up |
| | *Population dynamics in terms of individual-based fertility, nuptiality, and mortality* | Household demography model |
| | *Households interact with each other and form larger groups and settlements* | Household position model |
| Diet and nutrition | *Composition of foodstuffs consumed within a household and corresponding nutritional budget that regulates individual health* | Nutrition model and food consumption mechanisms |
| Food economy | *Processes involved in food production, beyond the procurement of raw foodstuffs, and distribution, storage, and exchange of foodstuffs* | Food storage model, exchange model |
| Decision-making | *Selection and revision of food production strategies at household level, particularly in terms of the triplet activity-conditions-investment, and other relevant aspects such as household position and the engagement with neighbors* | Mechanisms connecting labor investments, land use, diet satisfaction, and cooperation in food economy (several models) |

The Indus Village and all its parts are being progressively implemented, tested, and explored using NetLogo [33], a user-ready and well-established platform for simulation. Each module and version is also thoroughly documented with pseudocode diagrams and R markdown demonstrations [34,35]. To properly understand the model at each stage of complexity, we are performing extensive sensitivity analysis, part of which is presented in this article. To inform and test our hypotheses, we perform experiments designed to represent scenarios, particularly by varying certain key parameters or using alternative algorithmic designs.

More details on the TwoRains modeling program and the Indus Village overall design can be found in our previous publication [21] and the code and documentation are openly accessible under GNU GPLv3 licence in the dedicated GitHub repository [36].

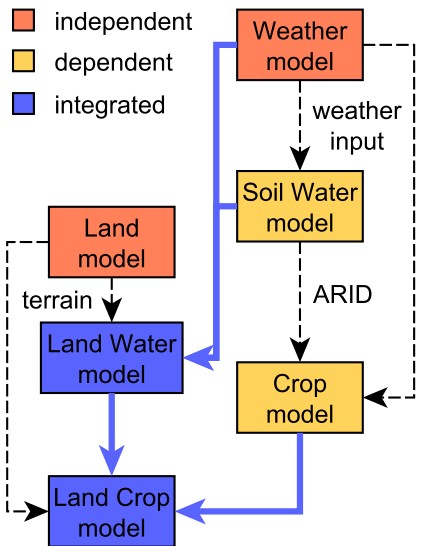

**Figure 1.** Road map of the models (submodels and integrated models) developed for the Indus Village model covered in this article.

### 2.2. Weather Model

The weather model is the first and most simple part of the Indus Village model. It was developed to generate daily values of solar radiation, temperature (average, maximum, and minimum), precipitation, and reference evapotranspiration; all of which are used by the soil water and crop models. The design goal was to create a general model that was controllable by meaningful parameters, and was also able to reproduce two rain seasons spaced by two dry seasons.

After a period of exploration, we defined the weather model as a set of parametric equations, based on two general equation-based models. One describes an annual sinusoidal curve approximating the seasonal variation of both solar radiation and temperature, while adding controlled daily fluctuation. $CO_2$ atmospheric concentration, an input of the crop model, is also generated using the same approach, but is deemed not relevant to the conditions registered for the IC, and thus is not used in the integrated models (see Section 2.4). Reproducing the much more irregular pattern of daily precipitation presents a harder challenge. Our solution was to first approximate the annual cumulative precipitation using a double logistic curve, which is then broken down into steps. Each slope sets the boundaries of a rainy season, and each step represents a distinctive rainy day. The annual cumulative curve is then converted into daily increments, which are scaled according to a predefined annual sum. Daily reference evapotranspiration is calculated using FAO-56 Penman–Monteith approach with constants set to the grass reference and estimating vapor pressure deficit from daily maximum and minimum temperature [37,38]. To reproduce the variance present in samples of multiple years, the model sets all parameter values at the beginning of each year stochastically using a normal distribution (i.e., the high-level parameters or *hyperparameters* are the mean and standard deviation of each low-level parameter).

We believe that the current model fulfills its purpose while balancing the number of parameters and assumptions with intelligibility and computational costs. Certainly, there are caveats and margin for improvement. For instance, there is no interaction between precipitation, at one side, and solar radiation and temperature, on the other, which are known to exist. Added to the materials already available [36], the in-depth description and parameter sensitivity analysis of the weather model will be delivered shortly as a separate article.

### 2.3. Soil Water Model

The soil water model is next in simplicity to the weather model. It describes the flow, retention, and drainage of water through the soil of land units. This model is an adapted version of the ARID soil water model, which, thankfully, has already been fully implemented in R and documented [39] (pp. 24–28, 138–144). It is a relatively simple model that calculates the daily water content of a single homogeneous layer of soil, determined by a given rooting depth, and is thus dependent on the vegetation considered. It requires the daily water input, including precipitation, and temperature, which is used to calculate the daily reference evapotranspiration. The targeted outcome is the Agricultural Reference Index of Drought (ARID) [40], which is a coefficient expressing the level of water stress in the soil in a scale typically ranging between 0 (none) and 1 (maximum stress, though the value can be higher if soil water level is lower than the plant wilting point). The soil water model was implemented in NetLogo together with the weather model so that it could remain independent of any given dataset.

### 2.4. Crop Model

Since its inception, the Indus Village was designed to represent aspects in multiple fields and disciplines. The ambition carried, among other challenges, the risk of exceeding a manageable level of complexity. Regarding crop modeling, many platforms have been developed by a large community along the last three to four decades [41,42]. These platforms were initially deemed as a potential pool of assets for our modeling program, as it has been in similar experiences in the past [25]. However, all platforms identified were found to be either restrictive in use, poorly documented, or too complex to be mastered at the developer level within the time frame and resources of TwoRains. Still, the main obstacle in adopting these platforms was their crop-specific approach, which would require either using multiple crop models as black boxes (methodologically undesirable) or reproducing each of them in NetLogo, so they could be fully integrated with other Indus Village submodels (technically unfeasible).

With this panorama in mind, we opted to use an adapted version of the SIMPLE crop model [43], which simplifies crop dynamics enough to easily allow for multiple crops, without dealing with multiple models and a high number of parameters. SIMPLE also integrates the ARID soil water model, which greatly facilitated the task of adapting it to the Indus Village framework. However, SIMPLE does not account for certain aspects that are integral parts of other crop modeling platforms and which are potentially important for representing ancient agricultural systems (e.g., soil fertility or N and C cycles, salinity, and root and leaf growing stages). Therefore, the adoption of SIMPLE should be considered a compromise and not a final solution.

The model consists of calculating the daily biomass based on growth stages defined in thermal time (i.e., cumulative daily temperature) and the modulation of the effective radiation use efficiency by temperature, $CO_2$ atmospheric concentrations, and soil water availability. Once a crop is sown, growth is triggered by daily mean temperatures higher than a threshold minimum ($T_{base}$) and is interrupted by harvesting. Yield ($g \cdot m^{-2}$) is derived as a fraction of the crop biomass at harvest, or harvest index ($HI$), unless thermal time passed is less than that required for maturity ($T_{sum}$), in which case crops are considered failed (i.e., no yield).

SIMPLE was adapted to NetLogo first as a base version, where the growth of each crop is simulated separately under the same fixed soil conditions (i.e., as if it were in the same land unit). Second and third versions were created to represent, respectively, spatial diversity (crops growing in different land units) and multi-cropping (crops growing in shares of the same land unit across a variety of land units). The latter was the one to be finally incorporated in the land crop model. Even though implemented in the weather and crop models, $CO_2$ concentrations were not included in the integrated models, given that $CO_2$ levels during the development of the IC were much lower than the threshold of effect defined by the model (i.e., 350 ppm).

The crop model does not factor in the economic side of agricultural systems, which forcibly include the mobilization and investment of workforce and other resources, as well as the social recognition of land use rights. Instead, as with most crop models, we must assume that the farming agency behind crop cultivation invests enough, so that all crops can be sown, maintained, and harvested successfully, according to the constraints covered in the model. However, a broader economic mechanism that integrates the factors above will make the core of the Indus Village [21,36].

Crop Selection and Parametrization

Following the SIMPLE approach, crop diversity takes form in the configuration of a relatively small number of parameters (Table 2). These are set in an external file ("CropTable.csv") and imported during initialization. Crop selection and parametrization is not a fixed aspect of the model and can be easily changed by editing this external file. The crops selected for the experiments presented in this article are limited to six cereals. However, the crop model inherits the flexibility of SIMPLE and could (and will) be used with other crops identified in IC contexts (e.g., pulses). Additionally, the crops selected are not controlled by empirical data on ancient cultivars of the Indus River Basin. Instead, they represent archetypes of cereal crops identified so far in the archaeological record, which might have had a significant role in throughout the Mature Harappan [13]. These crops were also selected to cover a range of meaningful parameter diversity within SIMPLE's definitions. Even though not calibrated, values are informed by crop modeling systems and crop-specific studies [41–46].

Within the scope of Indus Village, parametrization focused mainly on representing a set of cereal crops that differ in growing season (winter or *rabi*: wheat and barley; summer before monsoon or *zaid*: proso millet; summer during/after monsoon or *kharif*: rice and pearl millet) and water stress sensitivity (low: millets; medium: winter cereals; high: rice), while also exploring variations in thermal constraints and base radiation use efficiency (*RUE*).

The key role of the crop model in Indus Village is to mediate between ARID during a crop's growing season and the yield harvested. By design, the relationship between ARID and yield for a given crop is directly controlled by the water stress sensitivity or sensitivity of *RUE* to ARID ($S_{water}$). If the crop's $S_{water}$ is zero, there is no relationship and the crop is virtually unaffected by drought. Higher values of this parameter indicate some level of sensitivity to ARID, meaning that plants will be able to metabolize less solar radiation, consequently growing more slowly. Given the model simplicity, crops with considerably high water sensitivity (e.g., rice) will only reach the majority of their potential yield if ARID is kept close to minimum.

*2.5. Land Model*

The land model is a procedural generation engine that produces unique parametric terrains made of discrete land units. Unlike the weather and soil water models, its outcome is a single static set of spatial variables (multi-layer map), which can be exported (CSV) and later used as the initial state of land units in any of the integrated versions of the Indus Village model. The land model parameters can be configured purposefully to generate stochastic variations within a specific range of possibilities. All terrains included in this article were generated with a specific setting, aiming at reproducing the types of geomorphological conditions estimated for the sites in Haryana, namely, mainly alluvial plains that can vary from marshes and aquatic ecosystems, surrounding perennial or seasonal streams, to shrublands with dry sandy soils [47].

The first step in the land model is to set the elevation of land units. The sequence starts by sculpting discrete landform features using agents performing random walks that raise or depress elevation in their path. Elevation is then smoothed and perturbed with noise. Next, two algorithms are applied to generate map-wide slopes: one that inclines the horizontal plane in x (west–east) and (north–south); and another that forces the terrain

into either a valley or a ridge with a general north–south orientation. Although stochastic, the final outcome corresponds to a specific set of parameter values and a random number generator seed.

**Table 2.** Crop model parameters' values used to configure crops in all experiments [1].

| | Cultivar | | | | | | Species | | | | | | |
|---|---|---|---|---|---|---|---|---|---|---|---|---|---|
| **Crop** [2] | $T_{sum}$ | $HI$ | $I_{50A}$ | $I_{50B}$ | $T_{base}$ | $T_{opt}$ | $RUE$ | $I_{50maxH}$ | $I_{50maxW}$ | $T_{heat}$ | $T_{ext}$ | $S_{water}$ | $z$ [3] |
| proso millet | 1328 | 0.29 | 157.3 | 96.75 | 0 | 18 | 3 | 100 | 5 | 34 | 45 | 0.05 | 1000 |
| pearl millet | 1220 | 0.25 | 245 | 120 | 10 | 33 | 1.9 | 100 | 5 | 35 | 47 | 0.05 | 1000 |
| rice | 2300 | 0.47 | 850 | 200 | 9 | 26 | 1.24 | 100 | 10 | 34 | 50 | 1 | 400 |
| barley | 1762 | 0.42 | 350 | 170 | 0 | 15 | 1.24 | 100 | 20 | 34 | 45 | 0.4 | 1000 |
| wheat 1 | 2200 | 0.36 | 480 | 200 | 0 | 15 | 1.24 | 100 | 25 | 34 | 45 | 0.4 | 1000 |
| wheat 2 | 2150 | 0.34 | 280 | 50 | 0 | 15 | 1.24 | 100 | 25 | 34 | 45 | 0.4 | 1000 |

[1] Excluding $S_{CO_2}$ and suggested sowing and harvest day of year, which are present in "CropTable.csv". [2] Proso millet: *Panicum miliaceum L.*, hypothetical variety; pearl millet: *Pennisetum glaucum*, hypothetical variety; rice: *Oryza sativa*, IR72; barley: *Hordeum vulgare*, hypothetical variety; wheat 1 and 2: *Triticum aestivum*, Yecora Rojo and Batten, respectively. [3] Estimated values. Sources: for wheat and rice, Table 1a in [43]; for barley, hypothetical values broadly informed by DSSAT v4.7 source files and documentation [41] and APSIM v7.10 Crop Module Documentation [42] (RUE was kept equal to wheat and rice); for pearl millet, $HI$, $T_{base}$, $T_{opt}$, $RUE$, $T_{heat}$, and $T_{ext}$ were obtained in or informed by [44]; for proso millet, all parameters were informed by [45,46]; $z$, as well as sowing and harvest dates, were set based on [43], FAO-Ecocrop database (deprecated) and FAO-Land and water crop Information database (http://www.fao.org/land-water/databases-and-software/crop-information/; accessed on 10 January 2022). $T_{sum}$: Cumulative temperature requirement from sowing to maturity (°C d). $HI$: Potential harvest index. $I_{50A}$: Cumulative temperature requirement for leaf area development to intercept 50% of radiation (°C d). $I_{50B}$: Cumulative temperature till maturity to reach 50% radiation interception due to leaf senescence (°C d). $T_{base}$: Base temperature for phenology development and growth (°C). $T_{opt}$: Optimal temperature for biomass growth (°C). $RUE$: Radiation use efficiency (above ground only and without respiration) (g MJ$^{-1}$ m$^{-2}$). $I_{50maxH}$: The maximum daily reduction in I50B due to heat stress (°C d). $I_{50maxW}$: The maximum daily reduction in I50B due to drought stress (°C d). $T_{heat}$: Threshold temperature to start accelerating senescence from heat stress (°C). $T_{ext}$: The extreme temperature threshold when RUE becomes 0 due to heat stress (°C). $S_{water}$: Sensitivity of RUE (or harvest index) to drought stress (ARID index). $z$[3]: Maximum root zone depth (mm).

Once elevation has been established, the model calculates the direction of flow for each land unit, following the approach of Jenson and Domingue [48]. This direction always points to the lowest of up to eight neighbors in the grid (Moore neighborhood) or towards the edges of the map. Land units that are the lowest among their neighbors, also called sinks, are "filled up" or raised following Huang and Lee [49], until there are only sinks present at the edges of the map. With flow directions, the model can then calculate a measure of flow accumulation in each land unit, expressing how many land units flow directly or indirectly to it. Modifying this calculation, an arbitrarily large flow accumulation (controlled as the parameter `riverAccumulationAtStart`) is added to a land unit at the highest edge side and its flow direction is forced to point towards the inside of the grid. This added flow accumulation may or may not descend away from the origin, depending on the particular local topography. With this added feature, we aim at representing a passing stream or river without explicitly modeling its entire basin.

The model then derives, from flow accumulation and additional parameters, the soil depth and the relative proportion of sand, silt, and clay particle fractions. The model operates under the assumption that flow accumulation is indicative of past erosion. Overall, soil depth and the percentages of silt and clay increase downhill, while sand fraction increases uphill. Texture is classified using the US Department of Agriculture scheme, which can be related to tabulated data on soil water parameters.

A similar logic is applied to generate the composition of ecological communities within each land unit, which is expressed as relative proportions of grass, brush, and wood-type vegetation. In this case, however, we assume an order of ecological succession where wood components take precedence over brush, and brush over grass. The configuration of ecological communities generated by the land model was designed to serve as an initial

state in simulations and the climax of any further ecological succession, in case it changes due to perturbations. The immediate role of ecological communities in Indus Village is to inform a classification of cover types (i.e., woodland, shrubland, wood-grass, grassland, and desert), which are related to surface permeability (*run-off curve number*). The criteria used for separating cover types were inferred from the footnotes of Table 2.2 in [50]. Additionally, ecological communities form the base for estimating the subsistence value of the area not used for cultivating crops (i.e., used for herding, fishing, hunting, or gathering).

The NetLogo implementation of the land model is a fully-independent model that can generate terrains with a wide variety of characteristics, which can be explored in a controlled way through a set of meaningful parameters. Even though terrain generation can take less than a second in most machines, it includes an import/export functionality that can be used to generate entire terrain libraries, regardless of when these will be used in simulations. We also made efforts in facilitating the visualization of all generated aspects, particularly by adding multiple modes of patch coloring in the user interface.

*2.6. Land Water Model*

The first milestone towards a complete Indus Village is the integrated land water model, which combines the weather, soil water, and land models into a working independent model (Figure 1). The challenge was the integration of the terrains generated by the land model with the dynamics of the soil water model, which in turn already was seamlessly integrated to the weather model. The land model generates static terrain data, while the soil water model describes a dynamic process within each land unit separately. The combination of these allows the simulation of daily water input (precipitation, falling homogeneously throughout the terrain, and river water) being absorbed in each land unit and passed down the flow as run-off.

Most data generated by the land model are usable as parameter values of the soil water model, either directly or after the mediation of tabular data (e.g., a certain cover is associated with a value of surface permeability or *run-off curve number*). However, there are aspects that are not specified in either of these models, and thus required further design to make their integration possible. These are mainly three:

- *Estimation of soil water capacity thresholds or horizons according to soil texture*. Saturation and permanent wilting points are approximated using linear models factoring the percentage of clay in the soil (i.e., p_soil_%clay) based on data offered by the Soil and Water Assessment Tool (SWAT) Theoretical Documentation [51] (p. 149).
- *Redistribution of surface water (run-off and inundation)*. The run-off exchange algorithm combines the calculations in the soil water model with an algorithm in charge of redistribution, which was already implemented in the land model [48], and another to calculate run-off volume moving from one land unit to another, which was adapted from a model about flood impact [52].
- *Impact of water on ecological communities and cover*. When water input exceeds the capacity of the soil, surface water accumulates as an additional type of ecological component. Given a volume of surface water (mm·m$^{-2}$), the percentage of land unit area (1 ha or 10,000 m$^2$) covered by water is estimated assuming a bankfull width/depth ratio of 12 for all land units and a bankfull width equal to or less than land unit width (100 m). The width/depth ratio of 12 has been empirically identified as the most common value universally, and is a consequence of the physical processes governing the distribution of energy and resultant sediment transport [53]. The expansion of water surface is at the expense of dry ecological components (i.e., grass, brush, and wood), which are assumed to be affected evenly. When dry land is available, all ecological components grow following a logistic growth model, where the carrying capacity is proportional to the value in the initial ecological community configuration (given by the land model), minus the influence of water stress. Similar to the crop model, water stress over grass, brush, and wood is modeled in proportion to ARID and a water stress sensitivity coefficient. Both the intrinsic growth rate and

water stress sensitivity, as well as the maximum root depth used to calculate ARID, are controlled as parameters imported from the "ecologicalCommunityTable.csv".

More details on these solutions and the reasoning behind them can be found in the Indus Village repository [36] (at the directory "00-integrated-models/documentation/", or through the repository Wiki, at the entry "I1. Integrated Land Water model"). As in the case of the weather model, a full description and parameter sensitivity analysis of the land and land water models will be delivered as a separate article.

All submodels related to land units in the Indus Village model are designed to represent the conditions and processes involved in human subsistence. As mentioned, crop cultivation is considered to be the main subsistence activity, among the many alternatives available to the IC peoples. Consequently, most output variables of the integrated land water model feed directly into the crop model, which greatly facilitated the completion of the second integrated version.

*2.7. Land Crop Model*

The second modeling milestone is the integrated land crop model, which combines the integrated land water model with the crop model (Figure 1). Unlike the first milestone, the integration of components did not involve modeling additional mechanisms. The outputs of all former submodels feed directly into the required inputs of the crop model. The few adjustments required involve accounting crops as ecological components and further specifying the inputs related to crop choice and management in the absence of the rest of Indus Village submodels.

All crops growing in a land unit are considered together as an extra component of the unit's ecological community and cover type. Within the scheme set in the land water model, expanding surface water (i.e., inundation) continues to have preference over all other components. In turn, given its artificial nature, crop area has preference over all remaining components through active human intervention (i.e., clearing, deforesting, and weeding). The compositional approach to characterize ecological community and cover type in land units has an important implication, particularly for the potential advantages of low-intensity, multi-cropping agriculture. ARID and net solar radiation (an input of evapotranspiration and crop biomass growth) are calculated for the entire land unit, using weighted averages of albedo and maximum root depth of each component, both specified at component and crop levels in external tabular data ("albedoTable.csv", "ecologicalCommunityTable.csv", and "cropTable.csv"). This approach creates a stronger interrelationship between ecological components, including among crops. In terms of water availability, this setting generally penalizes the exposure of soils and awards the presence of components with deeper maximum rooting zone, whose roots help integrate a larger soil water layer.

Crop choice and management are represented in the land crop model as three variables, namely `selection`, `intensity`, and `frequency`. Selection (full name in code: `crop_selection`) is the list of crops to be simulated in land units, identified through the same unique names found in "cropTable.csv". Intensity (full name in code: `crop_intensity`) is the maximum percentage of the land unit area to be used by all crops selected, as farming plots are not equated to the model's land units. Frequency (full name in code: `p_crop_frequency`) is a list containing the percentage of effective intensity (`intensity` after discounting the effect of inundation) that is dedicated for each crop in `selection`. Human agency is not explicitly modeled in this integrated version. Therefore, to guarantee the operability of the land crop model, `selection` and `intensity` are set globally as parameters (i.e., all land units have the same set of crops occupying the same percentage of land unit area), while `frequency` is defined at the land unit level. `F_Solar_max`, which denotes the spacing between individual crop plants, is inherited from the crop model as a global parameter, but it is not explored in the experiments presented in this article. Furthermore, this temporary arrangement does not include sequential multi-cropping (i.e., crops being cultivated in different seasons in the same soil) [14] nor any other field rotation regime.

*2.8. Experimental Design*

As mentioned, Indus Village modular design allows for exploring each submodel and integrated models through separate simulation experiments. This article presents the results of six of such experiments, executed on the crop model (experiments 0, 1, and 2; Table 3) and the land crop model (experiments 3, 4, and 5; Tables 4 and 5). The specific configuration of these experiments are as follows:

- Experiment 0 serves as a baseline reference for further experiments by running the crop model for each of the six crops selected using empirical weather data instead of data simulated with the weather model. Data were obtained in the NASA POWER project Data Access Viewer and correspond to the location of the archaeological site of Rakhigarhi in Haryana, India, between 1 January 1984 and 31 December 2007 (24 years).
- Experiment 1 executed the crop model for 25 random number generator seeds and 30 years (total sample of 750 unique years), running in parallel for each of the six crops selected. The goal was to compare yield and water stress levels of crops under the default parameter configuration, which includes a specific configuration of the weather model parameters that aims at approximating the data used in experiment 0 (see calibration process in [54]).
- Experiment 2 is equivalent to experiment 1 in all aspects, except for varying stochastically (uniform probability distribution) the precipitation absolute total per year (`precipitation_yearlyMean`) and the winter/summer ratio of precipitation or the average plateau value of the cumulative curve of daily precipitation (`precipitation_dailyCum_plateauValue_yearlyMean`). This experiment aimed at exposing the effect of precipitation annual volume and seasonality on crop-specific ARID and yield.
- Experiment 3 executed the land crop model for 10 random number generator seeds and 5 years (total sample of 50 unique years), using five preselected terrains (2500 land units) and a default parameter configuration, both aimed at approximating past conditions in Haryana. `Frequency` was kept constant and homogeneous throughout all land units, while `intensity` was fixed at 50% throughout all runs.
- Experiment 4 is equivalent to experiment 3, except for varying stochastically the volume of river water inflow or, more specifically, the water stage increment ($mm \cdot m^{-2}$) per unit of flow accumulation at the river's starting land unit (`riverWaterPerFlow Accumulation`).
- Experiment 5 is again equivalent to experiment 3, except for varying stochastically (uniform probability distribution) the share of each crop in `frequency` within land units. The variation of crop frequencies aims at addressing crop choice factors by evidencing the effect of relative frequencies on total production per land unit (i.e., sum of all elements in `p_crop_totalYield`).

**Table 3.** Weather, soil water, and crop models parameters' values used in experiments 0 [1], 1, and 2.

| Submodel | Domain | Parameters | Unit | Default/Exp. 0, 1 | Exploration Range Exp. 2 |
|---|---|---|---|---|---|
| (all submodels) | time | year length in days | days | 365 | |
| Weather [2] | temperature | annualMaxAt2m | °C | 37.00 | |
| | | annualMinAt2m | °C | 12.80 | |
| | | meanDailyFluctuation | °C | 2.20 | |
| | | dailyLowerDeviation | °C | 6.80 | |
| | | dailyUpperDeviation | °C | 7.90 | |
| | solar | annualMax | MJ | 24.20 | |
| | | annualMin | MJ | 9.20 | |
| | | meanDailyFluctuation | MJ | 3.30 | |
| | $CO_2$ | annualMax | ppm | 245.00 | |
| | | annualMin | ppm | 255.00 | |
| | | meanDailyFluctuation | ppm | 1.00 | |
| | precipitation | yearlyMean | mm | 489.00 | 50–1000 |

**Table 3.** *Cont.*

| Submodel | Domain | Parameters | Unit | Default/Exp. 0, 1 | Exploration Range Exp. 2 |
|---|---|---|---|---|---|
| | | yearlySd | mm | 142.20 | |
| | (cum. sum) | nSamples | - | 200 | |
| | | maxSampleSize | days | 10 | |
| | | plateauValue_yearlyMean | mm·mm$^{-1}$ | 0.25 | 0.2–0.8 |
| | | plateauValue_yearlySd | mm·mm$^{-1}$ | 0.10 | |
| | | inflection1_yearlyMean | day of year | 40 | |
| | | inflection1_yearlySd | days | 5 | |
| | | rate1_yearlyMean | - | 0.07 | |
| | | rate1_yearlySd | - | 0.02 | |
| | | inflection2_yearlyMean | day of year | 240 | |
| | | inflection2_yearlySd | days | 20 | |
| | | rate2_yearlyMean | - | 0.08 | |
| | | rate2_yearlySd | - | 0.02 | |
| Soil water [3] | surface | elevation | m | 200 | |
| | | albedo | - | 0.23 | |
| | | runoff curve number (CN) | - | 65 | |
| | soil | drainage coefficient (DC) | - | 0.55 | |
| | | root zone depth (z) | mm | 400 | |
| | | field capacity (FC) | mm·mm$^{-1}$ | 0.21 | |
| | | water holding capacity (WHC) | mm·mm$^{-1}$ | 0.15 | |
| | | wilting point (WP) | mm·mm$^{-1}$ | 0.06 | |
| | | water uptake coefficient (MUF) | mm$^3$·mm$^3$ | 0.096 | |
| Crop | management | F_Solar_max | - | 0.95 | |

[1] Experiment 0 does not use the weather model. [2] Parameter names are simplified; in code, they all include domain-specific prefix (e.g., `temperature_annualMaxAt2m`). [3] Parameter names were kept in code in their abbreviated form, as they are in the original implementation in R. This is, however, not the case in the integrated versions, where their names are converted to their full name (e.g., CN is written as `p_soil_runoffCurveNumber`).

**Table 4.** Land model parameters' values corresponding to the terrains used in experiments 3, 4, and 5 [1].

| | | Terrain Random Seed | | | | |
|---|---|---|---|---|---|---|
| Domain | Parameters | 0 | 35 | 56 | 72 | 92 |
| elevation | algorithm-style | "C#" | "C#" | "C#" | "C#" | "C#" |
| | numDepressions | 7 | 10 | 3 | 4 | 6 |
| | numProtuberances | 4 | 6 | 9 | 4 | 8 |
| | numRanges | 66 | 43 | 38 | 64 | 92 |
| | numRifts | 88 | 97 | 13 | 63 | 42 |
| | rangeAggregation | 0.6458941 | 0.1113466 | 0.8133660 | 0.5878475 | 0.8563670 |
| | rangeHeight | 21.18274020 | 40.86174048 | 17.72232293 | 20.63073219 | 0.01770265 |
| | rangeLength | 1888 | 961 | 680 | 1279 | 1659 |
| | riftAggregation | 0.3834415 | 0.6789708 | 0.3916585 | 0.5006789 | 0.9385805 |
| | riftHeight | −48.183138 | −34.108734 | −43.865071 | −3.337115 | −22.063655 |
| | riftLength | 3090 | 959 | 1589 | 601 | 1686 |
| | featureAngleRange | 15.866848 | 1.374065 | 2.702853 | 28.400278 | 25.210475 |
| | noise | 3.958625 | 3.983587 | 1.630667 | 2.160736 | 3.588394 |
| | smoothingRadius | 3.464823 | 3.464823 | 3.464823 | 3.464823 | 3.464823 |
| | smoothStep | 1 | 1 | 1 | 1 | 1 |
| | valleyAxisInclination | 0.0871293 | 0.9890636 | 0.5495040 | 0.6342591 | 0.2892803 |
| | valleySlope | 0.0004043679 | 0.0037176302 | 0.0169948110 | 0.0116752657 | 0.0080676211 |
| | xSlope | 0.009255966 | 0.002138160 | 0.000962116 | 0.002474697 | 0.008352354 |
| | ySlope | 0.0007103606 | 0.0030363731 | 0.0064754508 | 0.0040414184 | 0.0044743707 |

**Table 4.** *Cont.*

| Domain | Parameters | Terrain Random Seed | | | | |
|---|---|---|---|---|---|---|
| | | **0** | **35** | **56** | **72** | **92** |
| flow | do-fill-sinks | true | true | true | true | true |
| | riverAccumulationAtStart [2] | 42,173,154 | 22,337,132 | 6,731,985 | 13,792,828 | 9,956,658 |
| soil | maxDepth | 561.0885 | 149.9677 | 491.1287 | 317.9024 | 464.0493 |
| | minDepth | 267.5030 | 105.5965 | 270.2213 | 206.9417 | 229.0143 |
| | depthNoise | 39.957929 | 25.789514 | 35.632002 | 8.418835 | 43.506614 |
| | formativeErosionRate | 2.334470 | 2.250253 | 1.322605 | 1.850611 | 2.567509 |
| | max%sand | 88.181043 | 43.658762 | 5.469936 | 84.602521 | 53.606080 |
| | min%sand | 46.147936 | 39.425832 | 3.567197 | 74.532106 | 17.096166 |
| | max%silt | 68.25092 | 34.55299 | 70.62584 | 76.87028 | 98.18986 |
| | min%silt | 11.82744 | 33.82258 | 32.77969 | 66.05339 | 77.34002 |
| | max%clay | 95.26008 | 97.08764 | 93.12337 | 11.49762 | 42.96254 |
| | min%clay | 14.33533 | 76.75279 | 74.57170 | 6.94711 | 35.25457 |
| | textureNoise | 5.218483 | 5.335943 | 9.901911 | 4.416463 | 3.734752 |
| ecology | woodFrequencyInflection | 39.26634 | 15.82286 | 35.38110 | 46.23942 | 19.50877 |
| | woodFrequencyRate | 0.00287898 | 0.04129579 | 0.09413197 | 0.02204882 | 0.06864615 |
| | brushFrequencyInflection | 13.686815 | 7.990782 | 19.641344 | 7.067255 | 14.663800 |
| | brushFrequencyRate | 0.04661503 | 0.08916186 | 0.05192546 | 0.06610931 | 0.07624894 |
| | grassFrequencyInflection | 2.0733097 | 0.4220637 | 4.8174187 | 3.5072296 | 0.9123332 |
| | grassFrequencyRate | 0.053911121 | 0.165866239 | 0.111039067 | 0.158674553 | 0.002558042 |

[1] Parameter stochastic sampling was performed by selecting type-of-experiment = "random" and setting the corresponding random seed in "land_05_withInitialEcologicalCommunity_Fig6version.nlogo" (value ranges and probability distributions, i.e., *hyperparameters*, are set in lines 302 to 363) [54]. [2] The amount of flow units added to a land unit at the edge of the map, expressed in land units (1 ha). To better scale this parameter, consider that the catchment area today of the entire Indus River Basin is approximately 116,500,000 ha, the Chenab River Basin in Pakistani Punjab is around 2,615,500 ha, and the Ghaggar River Basin in Haryana is 4,997,800 ha.

**Table 5.** Land crop model specific parameters' values used in experiments 3, 4, and 5.

| Domain | Parameters | Unit | Exploration Range | | |
|---|---|---|---|---|---|
| | | | **Default/Exp. 3** | **Exp. 4** | **Exp. 5** |
| terrain | seaLevelReferenceShift | m | −1000 | | |
| | riverWaterPerFlowAccumulation [1] | mm·m$^{-2}$ | $10^{-4}$ | $10^{-5}$–$10^{-3}$ | |
| | errorToleranceThreshold | mm·m$^{-2}$ | 1 | | |
| crops | selection | - | [all crops in Table 2] | | |
| | intensity | % of land unit | 50 | | |
| (land unit) | frequency | % of (effective) intensity (array of <selection> items) | 100/6 | | $C(0,1)$ [2] |

[1] Average river stage increment per flow accumulation at the river's starting land unit. Because there are many factors subtracting river flow (assuming that the catchment area is large enough, this parameter should be always very small; for the Indus Basin it would be approximately between $10^{-3}$ and $10^{-5}$ (near their mounths, this value would be around $5.66 \times 10^{-5}$ for the Indus (average discharge of 800.6 m$^3$/s and basin area of 2,615,500 ha) and $3.061 \times 10^{-4}$ for the Chenab (average discharge of 6600 m$^3$/s and basin area of 116,500,000 ha). [2] $C(0,1)$: uniform random distribution of each component in a composition (i.e., array with fixed sum).

### 3. Results

*3.1. Soil Water and Crop Yield in One-Land Unit Systems*

The results of experiments 0, 1, and 2, performed on the crop model, offer insight on the virtues and caveats of this approach in representing real-world patterns in daily

weather variables in Haryana, the corresponding ARID measurements, and, consequently, the effect on the yield of a variety of cereal crops.

There are considerable differences between the annual pattern described by weather variables in empirical data (experiment 0) and simulated data (experiment 1) (Figure 2). The characteristics of the reference dataset that contrast with the simulation data are as follows:

- *Solar radiation and temperature*: Springs are warmer and sunnier, summers are colder and less sunny, and autumns are warmer and sunnier. The distribution of both solar radiation and temperature is skewed and deformed when compared to the annual sinusoidal curve generated by the model. The annual maxima are reached one to two months before the summer solstice, and there is a considerable depression of both solar radiation and temperature due to the incidence of the monsoon.
- *Precipitation*. The summer monsoon tends to start and end sooner, while the winter rainy season is generally less intense. Differences in precipitation indicate the difficulty of representing the nuances of the two rains pattern found in the region. Daily maxima are also considerably lower, though this is not only a problem of misrepresentation, but also an effect of the limited sample size in the modern data.

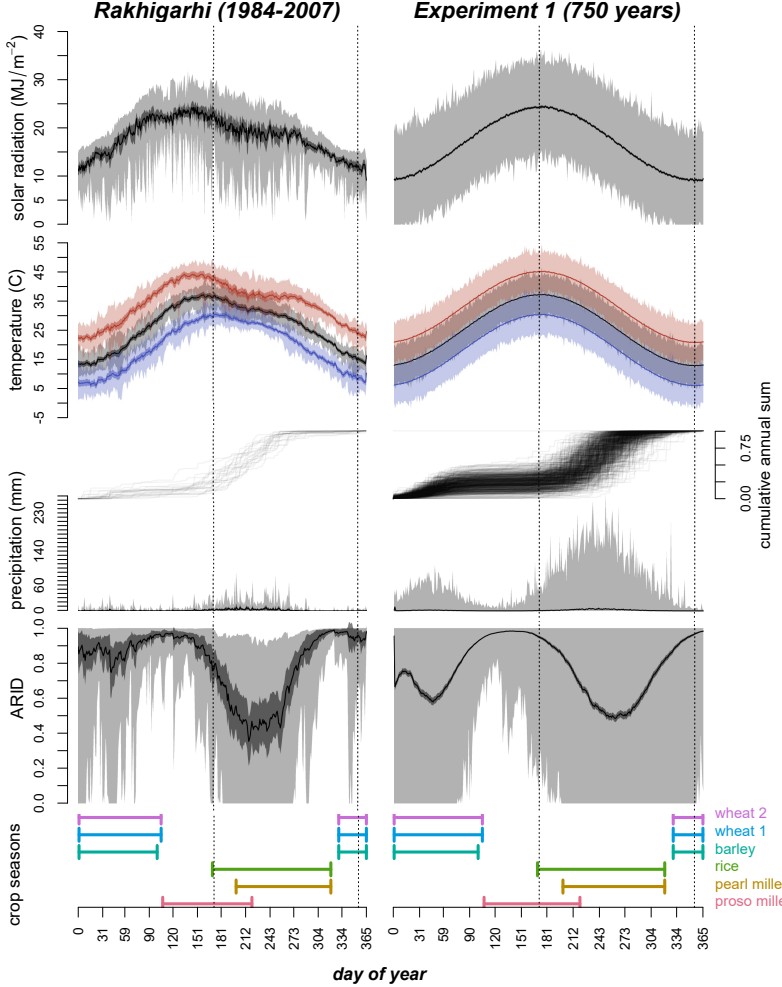

**Figure 2.** Summary statistics of weather variables and ARID per each day of year in (1) experiment 0, using data from Rakhigarhi, Haryana, India, between 1 January 1984 and 31 December 2007 (source: NASA POWER), and (2) experiment 1, corresponding to the simulation of 750 years (365 days). Means are solid lines, 95% Student's t-confidence interval are dark areas, and ranges between maximum and minimum are lighter areas. The growing seasons of the crops selected are plotted in relation to both datasets.

The mismatch between the empirical and simulated weather data is not unexpected, since there are several known processes that are not represented in the weather model and can affect all weather variables. Moreover, modeling precipitation in each year separately has the considerable limitation that the ending and beginning of consecutive calendar years are not seamless, which is particularly noticeable when parameter variation leads to early and short winter rain seasons (see in Figure 2, right, the small bump in the daily average at the beginning of the year). More refined parameter estimation techniques could render better results in terms of fitness to the modern dataset. However, better fitness might not be desirable considering that the weather patterns targeted refer to centuries of variation during the development of the IC, and are, for now, still largely unknown. Further investigation of the differences between real and simulated weather variables will be addressed in detail in a future publication dedicated to the weather model.

The calculation of ARID alone helps better understand the constraints of weather seasonality in Haryana. The so-called "two rains" pattern (i.e., two rainy seasons alternated with two dry seasons) contrasts with most other regions of the world, where precipitation is distributed more evenly throughout the year, irrespective of annual totals. More interestingly, results show the effect of the seasonal variation in daily temperature on how ARID reacts to precipitation. For instance, a relatively small amount of rain in winter can alleviate water stress significantly because of the lower levels of evaporation.

The highly seasonal nature of ARID in a context such as Haryana warns us about the dangers of searching for correlations between crop yield and annual mean ARID or other similar aggregate indicators. The large sample of year-series generated in experiment 1 demonstrates that there can be significant differences between the annual mean ARID and the means of specific growing seasons, particularly under highly seasonal weather, such as in Haryana (Figure 3; see one-way ANOVA and *post hoc* pairwise comparisons in [54]). According to the results of this experiment, *rabi* and *zaid* crops are more likely to suffer higher levels of soil water stress in comparison to the annual mean, while *kharif* crops would generally grow under lower ARID due to the summer monsoon. Despite the differences between simulated and empirical data, the particularities of ARID in each growing season still hold in experiment 0, with the possible exception of the *zaid* season, where ARID varies radically depending on the timing of the arrival of the monsoon.

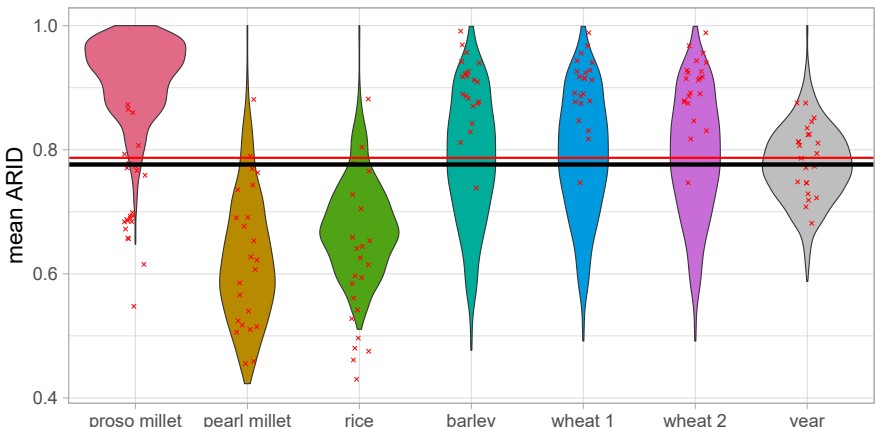

**Figure 3.** Mean ARID during each crop growing season and the entire harvest calendar year, as generated in experiment 0 (red points) and 1 (distribution shapes). The strong horizontal lines mark the average value of annual means for each dataset (red: experiment 0; black: experiment 1).

Under the framework set by the crop model, a crop's water stress sensitivity ($S\_water$) influences ARID during its growing season, which in turn modulates the rhythm of biomass growth and, consequently, the crop's yield. This interaction is proportional to both terms (i.e., $S_{water} \times ARID$, see [43], Equation (11)), which means that higher values of water sensitivity will amplify the effect of any temporary increase in ARID (i.e., dry spell). In

experiment 1, the variability in crop yield is related to this combined effect (Figure 4). Most notably, rice yield variance is shown to be negatively correlated with ARID during its growing season (Pearson's $r \approx -0.909$). This variance is attributed to the high $S\_water$ of this crop, as it is characterized in Table 2. Rice grows under the moisture of monsoon and post-monsoon conditions, so it is less likely to be exposed to high levels of ARID (Figure 3). Still, rice can be exposed to occasional peaks in ARID, whenever the high summer evaporation is unmatched by exceptionally lower or untimed precipitation. It is then when $S\_water$ assumes a key role for this crop's yield.

In contrast, proso millet exemplifies a different source of yield inconsistency. Being a C4 plant with a shorter growing cycle and parametrized with a distinctively high radiation use efficiency ($RUE$), this crop can take more benefit from the summer peak in solar radiation and temperature, producing the highest yield of the six selected crops. However, proso millet also presented the largest inconsistency in yield, exactly because it endures the hottest months of the year. Its growing season displays higher $ARID$ and, more importantly considering it has a low $S\_water$, higher daily temperature maxima. Days with such high maximum temperature are more likely to cross the crop's thresholds of tolerance to heat ($T_{heat}$, after which biomass growth declines linearly, and $T_{ext}$, where growth stops completely. While proso millet yield reaction to mean $ARID$ during growing season is negligible (Pearson's $r \approx 0.02$), its response to the mean daily maximum temperature of the period is considerable (Pearson's $r \approx -0.405$). Pearl millet, the most similar crop within the selection, is also more sensitive to mean daily maximum temperature, yet it is affected by it as a positive factor (Pearson's $r \approx 0.22$), given its growing season being set later in the year. Even though both proso millet and rice yields are more variable than those of the other crops selected, the variance in rice is more alarming from the perspective of a small producer, since bad years will more likely produce yields closer to zero.

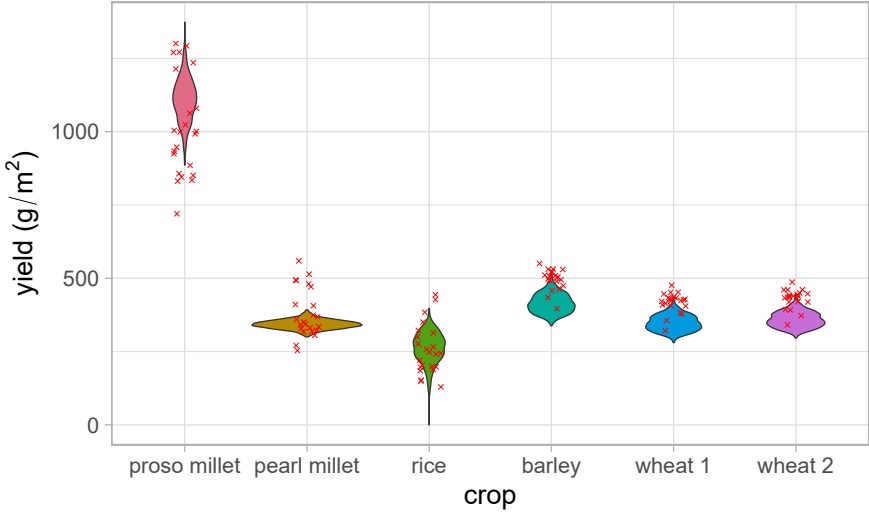

**Figure 4.** Harvested yield per crop in experiment 1.

The results of experiment 2, varying both the annual sum and the winter–summer balance of precipitation, further help to illustrate the complex implications of seasonality (Figure 5). While lower mean annual sums will generally correspond to higher mean $ARID$ during every growing season, this effect is distinctively less pronounced during the *zaid* season, when there is normally little precipitation under high evaporation conditions (Pearson's $r \approx -0.576$ for proso millet, compared to $-0.707$ for pearl millet, $-0.731$ for rice, and $-0.714$ for wheat).

This general negative relationship between precipitation annual sum and $ARID$ is demonstrated to be strongly modulated by the timing and distribution of precipitation, particularly in how much of it falls in the first and second rainy seasons, as they are

positioned on the Gregorian calendar ("winter" and "summer", respectively). This aspect is controlled by the parameter `precipitation_dailyCum_plateauValue_yearlyMean`, also explored in experiment 2, which defines the yearly mean of the proportion of precipitation falling in winter: i.e., a value of 0.3 indicates that, on average, 30% of the precipitation annual sum will fall as winter rain and 70%, as summer rain. Results displayed the expected relationships of this parameter with mean *ARID* during each growing season: positive for *kharif* crops (Pearson's $r \approx 0.427$ for rice, and 0.415 for pearl millet), negative for *rabi* crops (Pearson's $r \approx -0.374$ for barley, and $-0.375$ for wheat), and only slightly positive for *zaid* (Pearson's $r \approx 0.163$ for proso millet). Experiment 2 further confirms the implications for yield of each crop sensitivity to water stress. Both millets are greatly unresponsive to variations in precipitation, while the yield levels of the reminder crops have clear and proportional responses to these two precipitation parameters.

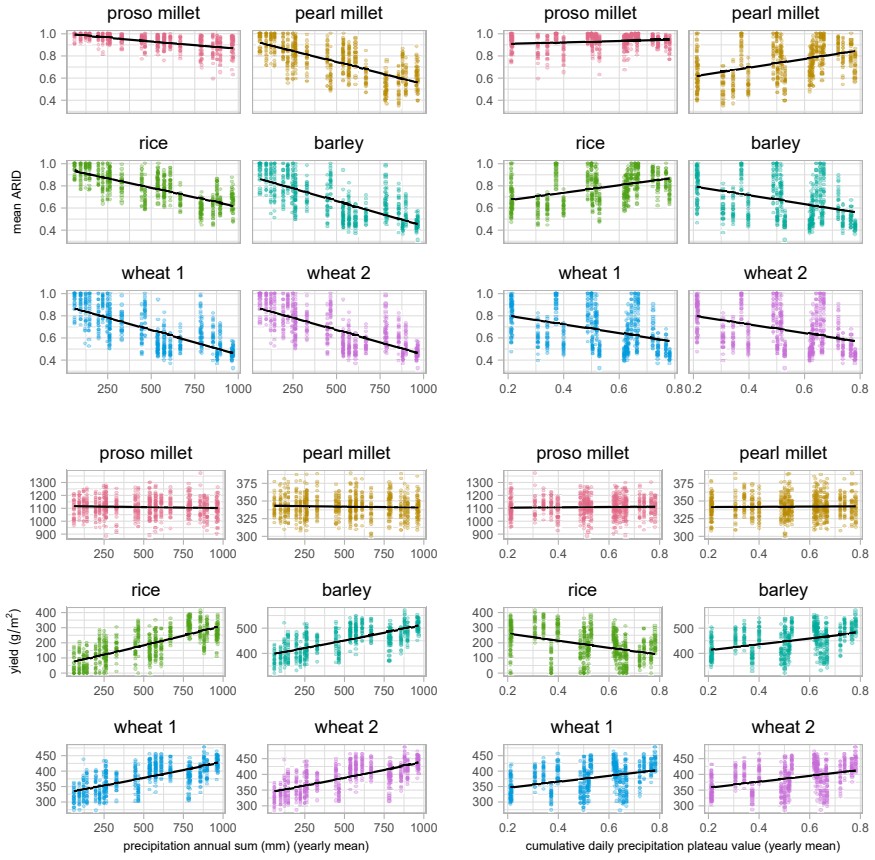

**Figure 5.** Mean growing season *ARID* and yield per crop versus the variation of two precipitation parameters: annual sum (yearly mean), and the plateau value of the cumulative curve of daily precipitation (i.e., the balance between winter and summer precipitation).

### 3.2. Soil Water and Crop Yield under Terrain-Like Systems

Experiments 3, 4, and 5, performed on the land crop model, were set to expand the points raised with the crop model by considering the consequences of spatial diversity within and between localities. For this, we preselected five terrains generated separately with the land model, using a single hyperparameter configuration and five random number generator seeds (0, 35, 56, 72, 92), which henceforth will be used as terrain names (Figure 6, Table 4). These five terrains were selected to represent a variety of topographic, soil, and ecological conditions, including whether a passing river makes it through the topography.

While all five terrains received additional flow accumulation in one land unit at the northern edge during the generation process, only terrains 35, 56, and 72 effectively contain a passing river due to their favorable relief. Regarding soil texture, terrains 0, 72, and 92

have soils with considerably higher sand fraction, dominated, respectively, by loamy sand (0), sandy clay (72), and loam (92); while terrains 35 and 56 have a greater percentage of silt and clay, classified overall as silty clay. Terrains 0 and 92, though dominated by sandy soils, contain spots with higher concentrations of silt and clay in and around drainage branches, in proportion to the local flow accumulation. Regarding the ecological communities covering terrains, wood and brush components are generally present along most drainage branches in all terrains, forming differentiated riparian communities characterized by woody vegetation (i.e., woodland and shrubland cover types). Land units positioned higher in the water flow chain have less wood and brush, and more grass, though falling still within the intermediate wood-grass cover type. The five terrains differ significantly in the overall distribution and balance of ecological components; e.g., terrain 0 is covered continuously by a single homogeneous brushy community, while 92 displays clear borders between woody and grassy communities.

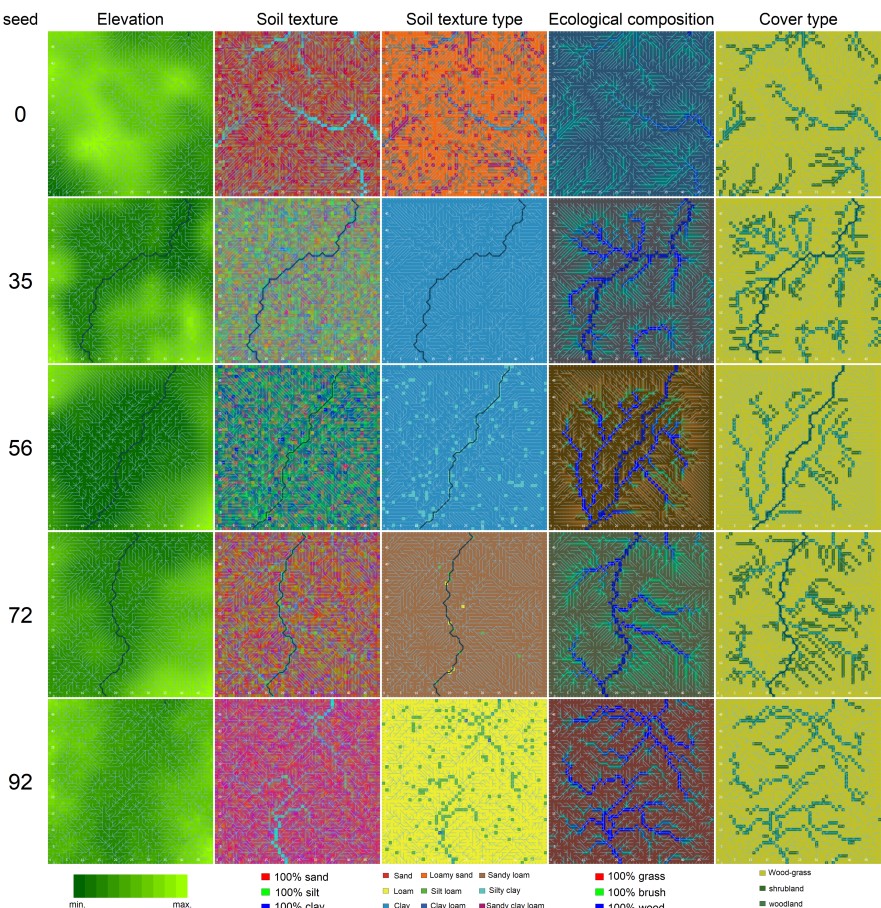

**Figure 6.** Terrains generated with the land model used in experiments 3, 4, and 5. Elevation, water flow, and soil texture composition and classification are displayed. Water flow is represented as blue links between land units, where thickness and color saturation are proportional to `flowAccumulation` at the origin. Soil texture types are derived using the USDA Textural Classification Chart [55]. Criteria for cover types are based on footnotes of Table 2.2 in [50].

In the land water and land crop models, terrain conditions combine with the dynamics set by the soil water model, given the input of the weather model, to produce daily spatial distributions of *ARID* that react differently to water inputs from precipitation and the passing river (Figure 7). Terrains with passing rivers (35, 56, 72) give way to the formation of larger inundation areas, particularly when the river path crosses flat reliefs. Land units in inundation areas will benefit from consistent lower levels of *ARID*, independent of the fluctuations in (local) precipitation. In no-rain days, landscapes are sharply divided

between such inundation areas ($ARID \rightarrow 0$) and the remaining land units ($ARID \rightarrow 1$). The advantage of inundation for terrestrial plants is conditional on whether, to what extent, and for how long land units are covered with free water, especially due to the influx of the passing river. Where the passing river is repealed (terrains 0 and 92), terrain conditions, such as flow distribution, soil type, and cover, are the factors further determining if at least some land units, particularly those down the flow chain, will be able to hold enough water to keep $ARID$ down during the time between rains.

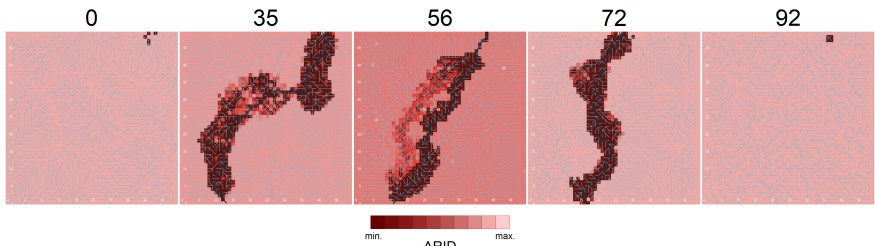

**Figure 7.** Example of $ARID$ spatial distribution for each terrain obtained with the land crop model at simulation step 365 (end of first year), a no-rain day, under the default parameter setting, and *randomSeed* = 0. Color saturation gradient represents the interval between case-wise minimum and maximum $ARID$.

In the land crop model, the spatial heterogeneity and subsequent land-water dynamics produced by terrain conditions directly influence crop productivity. The spatial distribution of crop yield (g/m$^2$) mirrors the one of $ARID$, because all other factors influencing crop growth and accounted for by the crop model are, in fact, homogeneous throughout all land units. Since yield spatial diversity is directly related to $ARID$, it too displays a clear distinction between those land units within inundation areas and the rest (Figure 8). The $ARID$-based spatial pattern of productivity holds for all selected crops, and therefore for all three growing seasons. Overall, inundation areas will offer the highest yields, while land positioned higher in the flow chain produces the least.

The contrast between inundation areas and the remaining land is also made clear when the overall distribution of yield per land unit is compared between terrains (Figure 9). The fewer high-yield values given by inundation areas tend to separate from the rest, similar to the rim of a long-necked bottle, which is clearer whenever there is a passing river (35, 56, 72). However, the impact of terrain conditions on a crop's yield is weighted by the combined effect of crop diversity and weather seasonality. As seen in the crop model results, $ARID$ will influence yield only in proportion to the crop's water stress sensitivity ($S_{water}$). This relationship still holds for the land crop model: compare the well-spread distribution of yield of land units in crops with low $S_{water}$ (i.e., millets) with the remaining crops, where it tends towards a binomial split. Furthermore, with $ARID$ being lowered by monsoon rains throughout all land units, the yields of rice are considerably more homogeneous than barley and wheat crops, discounting the splitting effect of inundation areas, which remains active throughout the year.

Experiment 4 allowed us to further explore the role of the inflow of a passing river by varying the volume of water corresponding to each flow accumulation unit added to model the river path (`riverWaterPerFlowAccumulation`). Considering that a flow accumulation unit represents the contribution of a land unit in surface water to a drainage basin, this parameter can be defined as the river stage contribution per land unit in the basin outside the terrain limits. Because this parameter is applied *after* the terrain is generated by the land model, it should be interpreted as representative of current conditions within the basin catchment area, and not during its longer geological and ecological history.

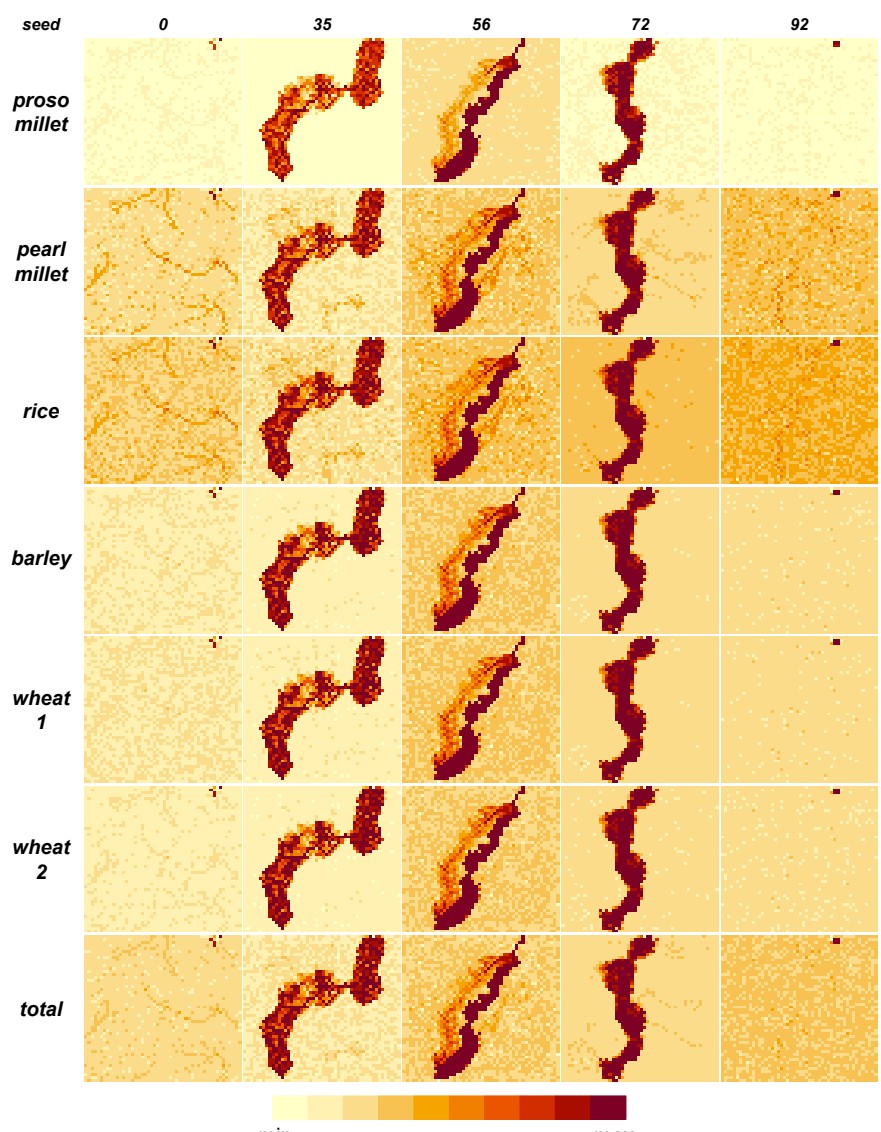

**Figure 8.** Mean yield spatial distribution (per crop and total) for each terrain in experiment 3. Color saturation gradient represents the interval between case-wise minimum and maximum yield.

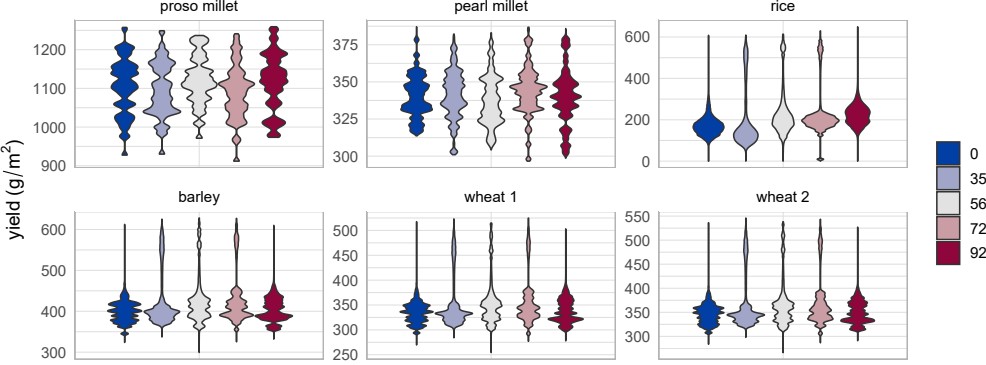

**Figure 9.** Yield of land units per crop and terrain in experiment 3.

The comparison of yield patterns under different values of `riverWaterPerFlowAccumulation` (Figure 10) confirms the positive effect of inundation areas, with increasing river flow causing a higher frequency of high-productivity land units, particularly for those crops with higher $S_{water}$. However, the experiment also raised a logical, yet easily forgettable,

point: larger river inflows may also cause larger portions of terrain to be fully covered by free water, thus decreasing the area available for any terrestrial vegetation, including crops. This detrimental effect is not negligible since the covered land units are forcibly the same that, due to the flow structure, tend to maintain lower *ARID* levels all year round. Increased river flow will benefit the gross yield of a crop the more it is sensitive to water stress; yet, those are also the crops that are less productive outside inundation areas, which makes them also the most prone to suffering the expanding free water. Therefore, there is a trade-off to the benefits of river inflow to a local crop economy, one that may pay off or not, depending on the extent of cultivation and which crops are prioritized.

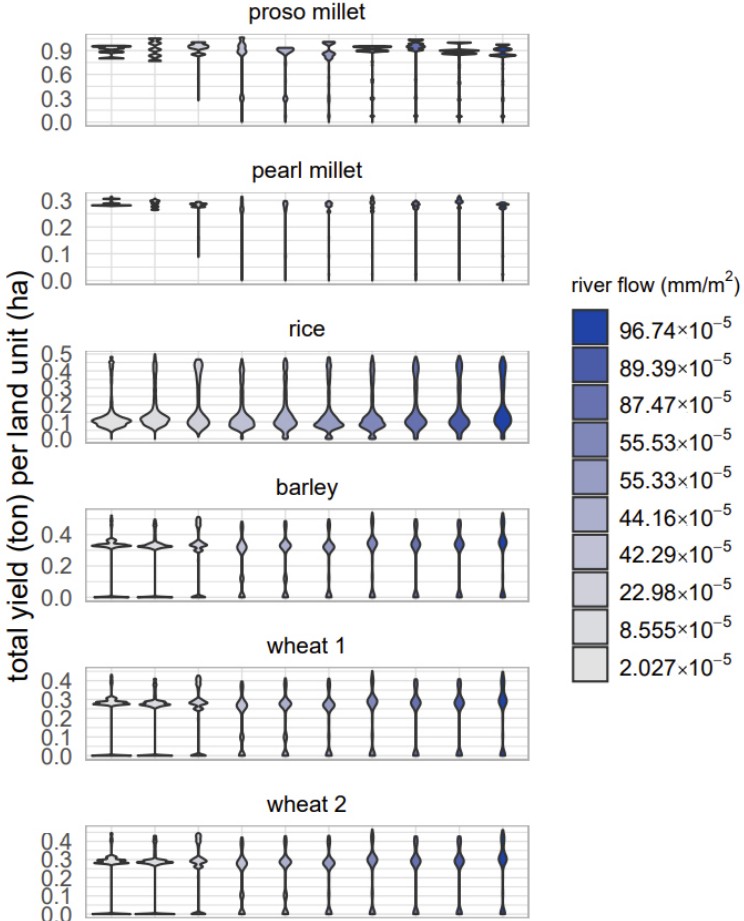

**Figure 10.** Production of land units per crop (`p_crop_totalYield`) for the different volumes of water per unit of flow accumulation in the river path (`riverWaterPerFlowAccumulation`) in terrain 35, as explored in experiment 4.

The importance of the availability of inundation areas for crops with higher water stress sensitivity is further highlighted by experiment 5 results, which expose the relationship between crop choice and the total production of a land unit. In experiments 3 and 4, crop frequency (`p_crop_frequency`) is effectively irrelevant when analyzing the relationship between *ARID* and yield (`p_crop_yield`), because the latter denotes productivity (mass per area; e.g., $g/m^2$) rather than production (mass; e.g., *ton*), thus being independent of the extent of each crop within each land unit. Moreover, if `intensity` and `frequency` are kept homogeneous and constant, as in experiments 3 and 4 (i.e., 50% of land units dedicated to crops, and 1/6 of that for each crop), the spatial distribution of production (i.e., sum of all elements in `p_crop_totalYield`) will always mirror the one of productivity (`p_crop_yield`). In experiment 5, however, every land unit has a particular configuration of `frequency` during each simulation run. The random variation of `frequency`, both between land units in the same terrain and between simulation runs in the same land unit, allows us

to examine how favoring one crop over others in particular land units can impact the total production of those land units and, consequently, the entire locality. We aim at exposing this relationship by measuring the correlations between the summed total yield and each crop's frequency. This indicator produces a noisy spatial pattern, given the stochastic nature of the exploration of `frequency` (Figure 11). Still, it reveals two points quite clearly:

- As a general rule, the total production of a land unit will increase with the more area that is used for proso millet, irrespective of its position, given the intrinsic high productivity and low sensitivity to water stress of this crop. Note how, in Figure 11, the range of correlation for this crop remains positive throughout all land units and terrains.

- Higher proportions of the more water-demanding crops will only tend to raise total production if the land unit is located in or around drainage branches and, particularly, inundation areas, in addition to having soil and cover conditions that facilitate lower *ARID*.

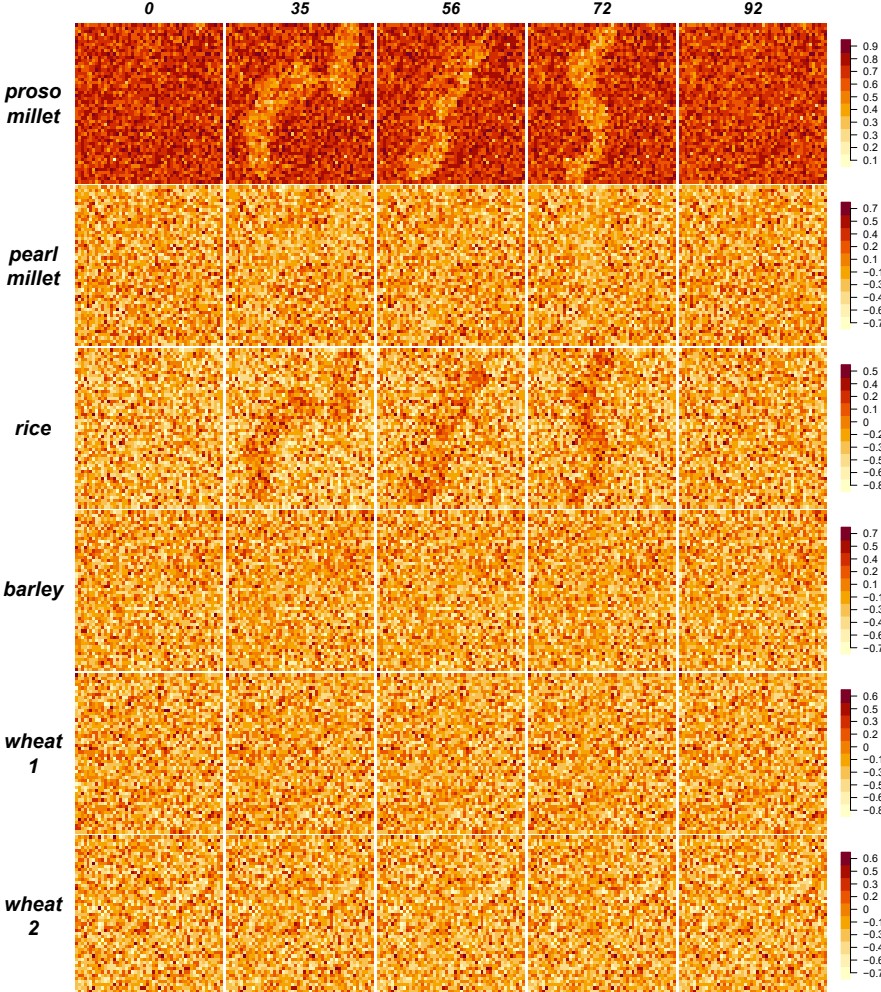

**Figure 11.** Correlations between crop frequencies and summed total yield per crop in each land unit of the five terrains, as generated in experiment 5. Color saturation gradient represents the interval between crop-wise minimum and maximum correlation values.

A direct consequence of this pattern is that an individual farmer agent, under the conditions experimented here, would be "playing safe" in terms of sheer total yield, whenever deciding to cultivate crops such as proso millet. However, if foodstuffs from all crops must be produced to a certain extent, the local economy would be more efficient whenever other, more sensitive, crops are limited to well-watered areas. In fact, from

a collective food economy perspective, the inundation areas formed by passing rivers (terrains 35, 56, and 72) offer a special opportunity for the cultivation of thirsty crops such as rice, making crops with lower $S_{water}$, such as proso and pearl millets, a less desirable option for the corresponding land units. Beyond the framework of our results, the argument for separating certain crops on the basis of their water stress sensitivity is further supported by evidence that sensitive crops such as rice can suffer and be out-competed by adjacent stands of millets [14,56].

## 4. Discussion and Conclusions

### 4.1. Crop Choice Dilemma

The progressive integration of the weather, soil water, land, and crop models, culminating in the land crop model, sets the context for a management dilemma that will have a key role in the broader Indus Village model. Such a dilemma (hereafter referred to as *crop choice dilemma*) is to be encountered by each farming agent (household) when choosing which crops to grow in a land unit, given the mandate of producing a certain amount of yield from a specific set of crops with diverse characteristics.

The six experiments described here were designed to help reveal or clarify the terms of the crop choice dilemma in the IC context, even if regarding only cereal crops, and they have widely met our expectations. According to our initial theoretical formulation [21] and the present results, the conditions leading to the crop choice dilemma are the following:

- **Crop selection**. There are a number of crops to be produced that are both available and required by society, given a broader economic and cultural context.
- **Crop diversity**. These crops are diverse as per their intrinsic biological traits, including those affecting when and how their growing cycle unfolds and, particularly, how sensitive these are to water stress.
- **Limited means of production and workforce**. Farming households, both separately and as a collective, count with a limited amount of land, labor, and other resources (e.g., tools, raw materials, fuel) to be invested in growing crops at any given time. Considering a pre-industrial agricultural system, extreme intensification of land use (i.e., near 100% of land unit area used for crops) is deemed neither desirable nor possible.
- **Seasonality**. Weather is highly variable within a year, with precipitation concentrating in seasons (two rainy seasons, two dry seasons).
- **Climate-driven variability**. There is a high interannual variation in weather, with great differences in the amount and distribution of precipitation between consecutive years.
- **Local environmental diversity**. Areas flooded by passing rivers and those down the water flow chain (e.g., dry channels, seasonal streams) represent a special type of local environment. This area is sharply separated from most of the remaining land, offering privileged crop-growing conditions with low, or no, water stress.
- **Regional environmental diversity**. The conditions shaping weather (seasonality and climate-driven variability), relief, soil properties, and ecology vary greatly between localities within the same cultural region, confronting the same cultural substratum to many potential contexts for the crop choice dilemma.

Our results, however, must be interpreted in light of the respective scopes and caveats of the crop and land crop models. We have admittedly focused on water availability as a limiting factor for crop growth, setting aside, at least for now, aspects likely to have some impact, such as soil fertility and salinity. Moreover, both crop and land crop models intentionally avoid many aspects related to the human side of an agricultural economy, which will be introduced with the remaining Indus Village submodels. For instance, despite their importance, workforce and other resources are not yet factored into the land crop model; they are, however, part of the planned development of the Indus Village model, building upon the experience of other models with similar scope and methodology [25]. By design, only a later integrated model will close the dynamic loop between crops and the human population (i.e., food processing, storage, and consumption; nutrition and its impact on demography; and the mobilization and investment of resources and labor).

Significant factors not yet addressed in our current framework are the interrelationships between crop cultivation and animal husbandry (e.g., crop biomass as fodder, manuring, animal traction for ploughing). Such connections have been documented as especially relevant in the case of millets [57–61].

### 4.2. Risk in the Crop Choice Dilemma

One of the most relevant dimensions of crop diversity to have a role in the crop choice dilemma is to what extent each cultivar is affected by fluctuations in the environment. Within the scope of our approach, sensitivity to soil water stress is the intrinsic factor driving most of the variability in productivity and thus determining the risk that investing in it entails. Solely according to this parameter (see $S_{water}$ in Table 2), we are able to rank crops from low to high risk (millets to winter cereals to rice), and correlate this with their yields under a wide variety of conditions. In light of our results, low-risk crops are either more predictable (pearl millet) or sustainably productive (proso millet) under virtually *any* regime of water availability offered in our experiments. In contrast, high-risk crops, such as rice and, in less measure, winter cereals, offer either low or inconsistent yields depending on if there is a systemic lack of water (i.e., low average annual precipitation, dry seasons, dry areas) or more temporary, unpredictable drying conditions. Despite the limited, estimative, and possibly anachronic (i.e., modern cereal cultivars) crop parametrization used in our experiments, the dynamics of the crop and land crop models suggest that this criterium could hold for differentiating most crop species and cultivars. Still, a question remains: Why did IC farmers continue to grow more sensitive/high risk crops?

The risk associated with a crop is weighted by farmers not only in relation to the sheer amount of biomass harvested, but also to its value per capita for consumption and exchange, as well as cultural reasons. A crop's value was likely grounded on multiple criteria, ranging from strict utilitarian calculation to traditional symbolism, including aspects such as flavor, appearance, familiarity, easiness to process, or integration in the local culinary tradition [13,62–67]. Many criteria for crop choice and food preferences have been indeed documented in modern societies practicing non-mechanized agriculture [61]. These are harder to identify archaeologically, but still have been a fertile field of research [64,66–72]. In this paper, we have chosen to focus on environmental factors in crop choice, specifically regarding cereals. However, IC archaeologists have been working to explore the social aspects of agricultural and food choices made by IC farmers and people [11–13,65,73–77], part of which is to be addressed in the remaining parts of the Indus Village model.

Independently of the criteria behind it, we expect that the value of a crop can potentially counterbalance the perceived associated risks. It could be, for instance, that the foodstuff and byproducts derived from a high-risk crop are valued to the point where a few good harvests compensate several bad ones, if certain economic conditions hold (e.g., there is a reliable exchange network, and the yield has high exchange value and can be stored long enough). One extreme scenario is where entire regions specialize in the production of one or few highly valued crops (*cash crops*), and obtain through trade all other goods required by its economy; a situation such as the one maintained by today's growing global demand for palm oil. A more likely scenario for the IC agriculture is that a wide variety of crops were demanded locally to a certain extent, and that the interregional exchange of foodstuffs, if and when it happened, had a more limited impact on crop choice. Altogether, we must assume that all crops identified to have been produced in IC sites were valued to the extent that perceived risks were deemed acceptable by farmers within their local conditions.

### 4.3. Risk Mitigation Strategies

Another side of the risk equation is that the risk itself can be reduced, to a point in which an otherwise risky crop can be safely cultivated, even if its value is not particularly high. There were many risk mitigation strategies available for prehistoric agriculture, each

to be manifested with their own risk-sensitive adaptive tactics, which could be adopted and transmitted differently on a village-by-village or even household-by-household basis [78].

Probably the most intuitive measure to minimize soil water stress in agriculture is to make more water available to crops. This *hydrophile* strategy can be performed by either sowing crops selectively in well-watered areas or mobilizing water into drier land (i.e., artificial irrigation). The results of the land crop model show quite clearly that this is a feasible option, as long as the water input remains within a certain range. Note that in all simulation experiments, river inflow was kept constant, making it a reliable water source independent of local precipitation. Under such conditions, we should expect that the hydrophile risk mitigation strategy is one of the main drivers of settlement patterns clustering in and around stable water sources, particularly, but not only, in arid and semiarid environments.

The preference for placing crops near water sources can be considered the extensification approach to the hydrophile risk mitigation strategy, and it is limited by the amount of free well-watered land and the economic and social costs of migration. Alternatively, artificial irrigation is the intensification side of the same coin, as it is an attempt to reduce risk by increasing the investment in the same location. It suffers from the limitation in both cultivable soil and extractable water available locally, and as such is also subject to diminishing returns. In addition, results from experiment 4 also suggest that external fluctuations bringing too little or too much water to the local system could undermine the effectiveness of this strategy in the long run.

Another, more holistic strategy to cope with risk is to spread the investment into multiple crops, environments, and agricultural land use practices. By improving and maintaining agrodiversity (namely, the biological, managerial, biophysical, and organizational diversity of the agricultural system), the farming agent can buffer the fluctuations in yield of individual crops while also safeguarding the future productivity of the entire food economy [79]. This *polyphile* risk mitigation strategy, of which multi-cropping is an essential part [14], has been documented in contemporary societies around the world and is demonstrably the most adaptive way of managing risk in agriculture. Far from being natural, agrodiversity is built up over many generations with the introduction of species from other habitats and practices that are molded by a long history of cultural exchanges and local adaptation. The polyphile strategy is not embraced as a form of optimizing agricultural production in a given year, i.e., by assigning the "correct" crop and practices to the "correct" environmental conditions; rather, the motivation for keeping agrodiversity comes exactly because there are no guarantees to which extent an optimal choice can be known in advance. Although not random, the return that can be expected from each combination of crops, practices, and environments is highly uncertain, even for the most experienced farmer, especially in contexts with so many dimensions of variability, such as the Indus River Basin. Moreover, and not less important, stewarding agrobiodiversity helps to conserve many plant and animal species that can be used as food, medicine, and other materials. Many of these species are not staple food crops, or even are strictly cultivated, but can play important roles as complementary or fallback sources of food (e.g., in the IC context, *Ziziphus sp.*) [80].

Although potentially leading to opposite practices, the hydrophile and polyphile risk mitigation strategies can be combined locally, either as household or village-wide traditions. A composite approach will give a different weight to each of these strategies, and would often include multiple rules of thumb specific to certain crops under certain conditions. Additionally, farmers can incorporate specific practices that reduce the overall risk by addressing other hazards, such as the use of manure and complementary crops (e.g., legumes) to replenish soil fertility. Other, more general risk mitigation strategies can also be added to the mix, such as common pooling, planning for a fallback surplus, or improving long-term storage infrastructure [81,82]. No matter how they are combined, the efficiency of all mitigation strategies relies on technology. Farmers following these strategies might have more or less success depending on how much labor and resources

they can mobilize, and how much knowledge about crops and their processing and storage was accumulated beforehand.

Even though it only considers a variety of six cereal crops, and has no representation of farming agents, the results of the land crop model already show how such a combination of risk mitigation strategies could work. Particularly, simulations in experiment 5 demonstrate that aggregate crop production in diverse and variable environments is more often higher if a mixed strategy is followed. The pull of cultivating in inundation areas is considerable, especially if thirsty crops are involved (necessity) and rivers are present (opportunity). Yet, diversity is clearly beneficial to explore most of the available crop niches, allowing to better exploit or buffer any environmental fluctuation.

### 4.4. Implications to IC Urbanization

IC farmers certainly relied on a long tradition of successful agriculture in the highly variable and diverse environment of the Indus River Basin. This tradition made use of a wide variety of crops, including species that originated within and outside the region, which contributed to the assembling of an approach to agriculture that was both cosmopolitan, regionally integrated, and locally adapted [13–15,17]. Agriculture before and during the urban phase of the IC (c. 2500–1900 BC) had deep connections to developments in the Fertile Crescent and the Iranian Plateau, with winter cereals (i.e., wheat and barley) occupying a central place [16], and it was intrinsically linked to a range of summer and winter cereal and pulse crops which were domesticated in different parts of South Asia [15].

However, the Indus region sits at the eastern limits of the distribution of winter precipitation triggered by the westerlies, which strongly limits the potential for rain-fed cultivation of winter cereals. Similar to in Southern Mesopotamia and Egypt, agricultural subsistence based on these crops could only be made sustainable by relying more on surface and underground water accumulations. If only accounting for this, we could prematurely conclude that IC farmers had relatively little room to maneuver outside the hydrophile risk mitigation strategy. This might indeed be the case in the arid Sindh, where the famous city of Mohenjo-daro was located. However, the scenario could be considerably more favorable and complex in the remaining parts of the region, which received either more winter precipitation (Pakistani Punjab), summer precipitation (Gujarat), or both (Eastern Punjab and Haryana) [10]. We expect that the opportunity offered by having both rainy seasons, as in Haryana, was not ignored by IC farmers, as preliminary archaeobotanical data seem to indicate that the dependence on winter cereals was balanced by the progressive investment in summer cereal crops (both *zaid* and *kharif*), yet never displacing winter cereals completely [83].

The cultivation of winter cereals has long been considered as one of the key factors in early urbanization in Afro-Eurasia, whether as an enabler or a booster of the surplus production sustaining a larger non-food-producing population [84]. Such consideration has also been raised specifically for the IC urbanism [16]. The relative specialization in these crops of early urban food economies has often been correlated to a greater emphasis on hydrophile risk mitigation strategies, with the increased practice of artificial irrigation and drainage of wetlands. We postulate that there is a tension exacerbated by urban growth regarding the crop choice dilemma: on one side, a hypothetical high exchange value of certain crops with water-dependent productivity, including winter cereals; and, on the other, the more sustainable approach of investing in multiple crop choice configurations simultaneously, being, however, less adapted to economies of scale. Even though long-term ecological dynamics will generally favor a more diverse system, social forces emerging with urbanization appear to have pushed local producers towards the opposite trend. In marginal lands, however, polyphile risk mitigation strategies are likely to have remained ever strong, given that mountains, deserts, and dense forests remain as sanctuaries of agrodiversity still today.

The different suitability of the hydrophile and polyphile risk mitigation strategies can also exist at a strictly local scale. The results of experiment 5 confirm this by showing that

each crop corresponds to one particular niche map or distribution of summed total yield. Out of our crop selection, the two millets are the safest choices for any given land unit because of their insensitivity to water stress. Crops such as these (we suspect most millets) would have been the best candidates for inclusion whenever farmers favored the polyphile strategy. In contrast, the remaining crops can significantly decrease the total yield of a plot if soil water is lacking. Prioritizing these other crops required a more intensive adoption of the hydrophile strategy, which locally is only sustainable for those households that have access to well-watered lands.

Although seemingly obvious from a bird's-eye and post-factum perspective, the diversity of local conditions and their variability through time pose a hard problem for farming decision-makers, assuming they do not have unrestricted access to land and knowledge. Even if there were open communication and some form of collective action to share risk and coordinate risk mitigation, we expect that each household was ultimately responsible for their choice and their members are the ones most affected by its success or failure. Moreover, we may conjecture that the crop choice and risk mitigation strategies of IC farmers were subjected not only to recurrent utilitarian revisions but also to strong traditional convictions transmitted generation by generation.

As already mentioned above (Section 4.3), we expect that the long-term agroecological dynamics will tend towards the prevalence of the polyphile strategy over the hydrophile strategy. However, this is assuming agriculture is strictly aimed at local subsistence (i.e., small-scale food production). At all scales where IC urbanization was manifested (cities, towns, specialized outposts), it likely involved an increased pressure in local food economies. Such pressure was most likely focused on cereals, pulses, and other foodstuffs that could be more easily processed, stored, and transported in substantial quantities; all properties which may have increased the value of the production of some high-risk crop regimes [16,18,65,85]. When exchange becomes an important factor in crop choice and risk assessment, the traditional balance between polyphile and hydrophile risk mitigation strategies could have tilted in favor of the latter. This would have implied a progressive shift of traditions from higher agrodiversity to higher risk-taking, with farming households, particularly in and around urban centers, becoming more and more dependent on regional distribution processes for obtaining foodstuffs and other goods.

If the hypothetical link between the hydrophile strategy and urbanization were to be confirmed, it would shed light on the apparent contradiction of IC urban centers being both resilient to high variability in local environments while also being vulnerable to climate change. A large-scale drying period, such as the one starting at c. 4.2 k BC, would have reduced not only local precipitation, but also the overall hydrological inflow in the entire region. As in many other early urban societies, the complexity and potential of the IC food economy would have been adaptive in respect to short-term, local environmental fluctuations, yet large urban centers were rendered unsustainable when the key element of their risk mitigation strategy (i.e., stable access to water) failed consistently and at a regional scale. In contrast, while also resilient to the uncertainty of local environments, the IC rural population resisted the consequences of climate change, as suggested by the continued occupation of the region's hinterlands. We hypothesize that such continuity would have been the consequence of their marginality within the urban IC food economy and thus their more autonomous drift towards the more adaptative polyphile strategy.

This explanation prompts a number of further research questions. For example, what would be the trajectories taken by our artificial IC villages if there is an increasing or decreasing trend in precipitation and river inflow? How important is the seasonality of these changes? Would the effectiveness of different combinations of risk mitigation strategies change during these trajectories? Would the frequency of the hydrophile and polyphile risk mitigation strategies in the local population change in response to the climate? These questions and the overall explanation outlined here will be explored further in our next steps, where we will make use of the other parts of the Indus Village model, and design experiments that represent climate change through a programmed change in the

hyperparameters controlling precipitation and river inflow, which are already present in the land crop model.

**Author Contributions:** Conceptualization, methodology, software, validation, formal analysis, investigation, visualization and data curation by A.A.; writing—original draft preparation, A.A. and C.A.P.; conceptualization, writing—review and editing, A.A., C.A.P., J.B., J.-P.B., A.G., J.R.W., M.C.U., N.W. and R.N.S.; resources, supervision, project administration, funding acquisition, R.N.S. and C.A.P. All authors have read and agreed to the published version of the manuscript.

**Funding:** This paper was made possible thanks to funding from the European Research Council (ERC) under the European Union's Horizon 2020 research and innovation program (grant agreement no 648609) for the TwoRains project (www.arch.cam.ac.uk/research/projects/tworains; accessed on 10 January 2022). TwoRains is a collaboration between researchers at the University of Cambridge and Banaras Hindu University in Varanasi that was begun as the Land, Water and Settlement project (www.arch.cam.ac.uk/research/projects/land-water-settlement/ (accessed on 10 January 2022)). This research has also benefited from support from the Global Challenges Research Fund's TIGR2ESS (Transforming India's Green Revolution by Research and Empowerment for Sustainable Food Supplies) Project, which was supported by the Biotechnology and Biological Sciences Research Council Grant Number BB/P027970/1. We would also like to acknowledge funding from the DST-UKIERI program and the McDonald Institute for Archaeological Research.

**Institutional Review Board Statement:** Not applicable.

**Informed Consent Statement:** Not applicable.

**Data Availability Statement:** All files related to this article are available as a public online repository at https://github.com/Andros-Spica/Quaternary_Angourakis-et-al-2022_Rproject (accessed on 10 January 2022) or at Zenodo [54]. These include a full Rmarkdown/HTML report on results, and the datasets and source files for models, figures, and statistical analyses.

**Acknowledgments:** Meteorological data used in Figure 3 were obtained from the NASA Langley Research Center (LaRC) POWER Project funded through the NASA Earth Science/Applied Science Program (https://power.larc.nasa.gov/; accessed on 10 January 2022).

**Conflicts of Interest:** The authors declare no conflict of interest.

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
