# Peer review of "Weather, Land and Crops in the Indus Village Model: A Simulation Framework for Crop Dynamics under Environmental Variability and Climate Change in the Indus Civilisation"

_quaternary, doi:10.3390/quat5020025_

Round 1
Reviewer 1 Report
Why Haryana was the sole state of investigation while the IC culture was settled on Punjab, Haryana, and Gujarat?
Some references of equivalent models may explain why this scale and why this resolution: are there fields of 100m*100?
All Indus Village sub models are designed and, at least partially, implemented: here the various sub models should be detailed (with/without what? And combinations?)
The methodology used could be structured through article references on SES ABM models:
Kohler T.A., Gumerman, G.J., Reynolds, R.G., 2005. Simulating ancient societies: computer modeling is helping unravel the archaeological mysteries of the American Southwest. Scientific American.
Kohler T.A., van der Leeuw, S.E., 2007. The model-based archaeology of socio-natural systems. Oxbow Books Ltd, Oxford, UK.
Barceló J.A., Del Castillo, F. (Eds.), Simulating Prehistoric and Ancient Worlds, Springer Computational Social Sciences. Springer.
Saqalli M., Vander Linden M. 2019. Introducing Qualitative and Social Science Factors in Archaeological Modelling. Simulating the Past Books Series. Springer Editions, New York, USA.
Pseudocode diagrams: UML or not?
A holistic model, covering aspects not in one but many fields and disciplines: sorry but no: holo- means explicitly covering all elements (that matter). Unless explicit parameters are used to explain and justify why some elements are negligible and so can be neglected, a model can be holistic if and only if all factors/disciplines are integrated, even in a simple matter. Her, the model is multidisciplinary and even transdisciplinary but not holo-disciplinary as no justification is provided to explain that the elements of the model are sufficient to explain its behavior. For instance, no rules on inheritance and intrafamily manpower are established. The model is mostly agro climatic and no manpower constraints is explicit meaning this s a maximalist model as a major constraint in farming systems is forgotten. Please correct
Some references (see above) are missing for showing the lack of multidisciplinary works. Please lower the exclusivity of this approach
Why the weather model did not follow a data based pattern such as an adapted WorldClim model? To avoid such gaps between theory and reality?
Most other regions of the world all eastern southwestern and central Africa or central South America meaning several millions of km² have the same pattern. The majority is based on one unimodal dry and rainy season: both are the majority of land masses on earth and the majority of the human population. . It is only not western.
For the climate: what about the normal accidents, i.e. droughts, meaning the inter annual and the intra annual variability and the spatial variability of rains?
Are you sure pearl millet & rice behave the same like indicated in figure 3?
Pearl millet do react to rainfall but also manure: see:
Affholder, F., 1997. Empirically modelling the interaction between intensification and climatic risk in semiarid regions. Field Crops Research 52, 79–93.
Akponikpé, P.B.I., Bielders, C.L., Gérard, B., Michels, K., 2007. Modeling integrated fertility management strategies for pearl millet in the Sahel, in: de Neve, S., Salomez, J., van den Bossche, A., Haneklaus, S., van Cleemput, O., Hofman, G., Schnug, E. (Eds.), .
Akponikpé, P.B.I., Minet, J., Gérard, B., Defourny, P., Bielders, C.L., 2011. Spatial fields’ dispersion as a farmer strategy to reduce agro-climatic risk at the household level in pearl millet-based systems in the Sahel: A modeling perspective. Agricultural and Forest Meteorology 151, 215–227.
De Rouw, A., 2004. Improving yields and reducing risks in pearl millet farming in the African Sahel. Agr. Sys. 81, 73–93.
Fatondji, D., 2002. Organic fertilizer decomposition, nutrient release and nutrient uptake by millet crop (Pennisetum glaucum (L.) R.Br.) in a traditional land rehabilitation technique (Zaï) in the Sahel. ZEF / ICRISAT, Niamey, Niger.
Fernandez-Rivera, S., Hiernaux, P., Williams, T.O., Turner, M.D., Schlect, E., Salla, A., Sangaré, M., Ayantunde, A.A., 2005. Nutritional constraints to grazing ruminants in the millet-cowpea-livestock farming system of the Sahel, in: Coping with Feed Scarcity. pp. 157–182.
Matthews, R.B., Pilbeam, C., 2005. Modelling the long-term productivity and soil fertility of maize/millet cropping systems in the mid-hills of Nepal. Agr.Eco.Env. 111, 119–139.
Climate variability, workforce, crop selection & diversity are finally described but in discussion. They should be in materials & methods even if not producing results yet. Indeed, one may see with difficulty the link between experiments and criteria proposed in discussion: where was the workforce before? So, either you make the link between experiments and these factors so they should be in M&M, either you propose them in discussion explaining where you found them with some references on manual farming systems for instance! They are not optional, especially the last two points which are intrinsically part of the models themselves. All are agronomic variations but is the real issue to prove they matter individually or to prove that they strongly affect the system altogether?
Manure is both a fertilizing and risk mitigating strategy and the IC was established in introduction that it was an agropastoral civilization where livestock is important. This very important component should be acknowledged as manpower as the most important constraints along water.
Association of cultures especially with legumes, the effect on fertility which does not reduce itself along crops are important missing points that could be understandable if and only if they are explicitly described in M& M and justified why they are neglected. I am on my point of view quite convinced that the cereal used is not the major point of a farming system but rather the complementary articulation of all crops, both in cycles, water, pests and fertility, and the cover of all needs (hydrocarbon, protein, textile and drugs)
What is the difference between agrodiversity and agrobiodiversity here?
Yet more provincial: what does it even mean for present time conceptions of agriculture and even more for prehistoric and historic times? As if they were fashionable agriculture? Please remove this useless comment
As a total counterexample of the dependence to water for an urbanized civilization of the hydrophile strategy (please insert the fact that polyphile is usually called diversification), the Sahelian kingdoms (Mali and Songhai) did rely on the Niger for transportation but not on irrigation, with a capital city of an attested population of over 10 000 inhabitants. Is that stated that non irrigated fields were not cultivable?
conclusion: very nice article but a need for reducing the slf attribution of the word holistic: the model is agronomic/agroclimatic and half of the constraints (ie social) aremissin) and CANNOT be neglected. Many crucial agronomic factors as association ofcultures and manure (on which the major part of the fertility and sustainibility of non modern farming systems rely on) are missing as well
Reviewer 2 Report
The authors of this paper are collaborating in designing and testing a series of agent-based models focusing on modeling communities and their development of early farming cultures in the Indus Valley. This paper introduces a set of partial models focusing on the growth of various crop types and relevant variables in the environment, e.g. topography, hydrology, soil-related factors as well as climate and the respective seasonal dynamics. The individual partial models themselves are already published and well documented. This paper therefore introduces, tests, and discusses the complex integration of them as a representation of a variable, not fully predictable and dynamic environment.
The basic question which is supposed to be addressed by the overarching Indus Village Model is the decline and finally the collapse of cultures in the Indus Valley in the Late Harappan Period after 1,900 BC. In order to answer this question, the environmental model needs to be supplemented by models for the behavioral strategies of the agents. However, the growth of a variety of crops, the presence of which is documented in the archaeological record, is considered as a crucial factor in this process which is largely determined by environmental dynamics.
The integration of the partial models is considered as a stepwise process. The authors designed a set of 6 experiments with increasing degree of integration. Climate dynamics in general and precipitation in particular are contrasted and compared with recent climate data in the first step. Such comparisons serve at least three purposes in this context: firstly, they validate the model results, secondly, they allow for calibration of crucial input factors, and finally, they allow to carry out sensitivity studies through systematic variation of single input factors.
The main output factor which is used to compare the performance of various types of crops is the yield, which directly depends on environmental dynamics, although the models themselves can easily generate further output variables.
The presentation of the results is exhaustive, careful and convincing. I particularly appreciate the discussion, in which the concept for integration as well as the results of the experiments are contextualized with archaeological evidence of the Harappan cultures.
In sum, the manuscript is carefully written and the presentation of the models and the integrative concept is convincing.
Two minor comments remain:
- Page 16, line 451: “the most yield” does not sound as elegant as the manuscript deserves. Please replace ‘most’ by ‘highest’.
- The scaling of the y-axis for yields in figures 4, 5 (lower part), 9, and 10 is highly variable and makes comparisons quite difficult. Please homogenize.
